# CASTLE: Regularization via Auxiliary Causal Graph Discovery

**Trent Kyono**[*]
University of California, Los Angeles
tmkyono@ucla.edu

**Yao Zhang**[*]
University of Cambridge
yz555@cam.ac.uk

**Mihaela van der Schaar**
University of Cambridge
University of California, Los Angeles
The Alan Turing Institute
mv472@cam.ac.uk

## Abstract

Regularization improves generalization of supervised models to out-of-sample data. Prior works have shown that prediction in the causal direction (effect from cause) results in lower testing error than the anti-causal direction. However, existing regularization methods are agnostic of causality. We introduce Causal Structure Learning (CASTLE) regularization and propose to regularize a neural network by jointly learning the causal relationships between variables. CASTLE learns the causal directed acyclical graph (DAG) as an adjacency matrix embedded in the neural network's input layers, thereby facilitating the discovery of optimal predictors. Furthermore, CASTLE efficiently reconstructs only the features in the causal DAG that have a causal neighbor, whereas reconstruction-based regularizers suboptimally reconstruct all input features. We provide a theoretical generalization bound for our approach and conduct experiments on a plethora of synthetic and real publicly available datasets demonstrating that CASTLE consistently leads to better out-of-sample predictions as compared to other popular benchmark regularizers.

## 1 Introduction

A primary concern of machine learning, and deep learning in particular, is generalization performance on out-of-sample data. Over-parameterized deep networks efficiently learn complex models and are, therefore, susceptible to overfit to training data. Common regularization techniques to mitigate overfitting include data augmentation [1, 2], dropout [3, 4, 5], adversarial training [6], label smoothing [7], and layer-wise strategies [8, 9, 10] to name a few. However, these methods are agnostic of the causal relationships between variables limiting their potential to identify optimal predictors based on graphical topology, such as the causal parents of the target variable. An alternative approach to regularization leverages supervised reconstruction, which has been proven theoretically and demonstrated empirically to improve generalization performance by obligating hidden bottleneck layers to reconstruct input features [11, 12]. However, supervised auto-encoders suboptimally reconstruct all features, including those without causal neighbors, i.e., adjacent cause or effect nodes. Naively reconstructing these variables does not improve regularization and representation learning for the predictive model. In some cases, it may be harmful to generalization performance, e.g., reconstructing a random noise variable.

---

[*]Equal contribution

Although causality has been a topic of research for decades, only recently has cause and effect relationships been incorporated into machine learning methodologies and research. Recently, researchers at the confluence of machine learning and causal modeling have advanced causal discovery [13, 14], causal inference [15, 16], model explainability [17], domain adaptation [18, 19, 20] and transfer learning [21] among countless others. The existing synergy between these two disciplines has been recognized for some time [22], and recent work suggests that causality can improve and complement machine learning regularization [23, 24, 25]. Furthermore, many recent causal works have demonstrated and acknowledged the optimality of predicting in the causal direction, i.e., predicting effect from cause, which results in less test error than predicting in the anti-causal direction [21, 26, 27, 28].

**Contributions.** In this work, we introduce a novel regularization method called CASTLE (CAusal STructure LEarning) regularization. CASTLE regularization uses causal graph discovery as an auxiliary task when training a supervised model to improve the generalization performance of the primary prediction task. Specifically, CASTLE learns the causal directed acyclical graph (DAG) under continuous optimization as an adjacency matrix embedded in a feed-forward neural network's input layers. By jointly learning the causal graph, CASTLE can surpass the benefits provided by feature selection regularizers by identifying optimal predictors, such as the target variable's causal parents. Additionally, CASTLE further improves upon auto-encoder-based regularization [12] by reconstructing only the input features that have neighbors (adjacent nodes) in the causal graph. Regularization of a predictive model to satisfy the causal relationships among feature and target variables effectively guide the model towards the direction of better out-of-sample generalization guarantees. We provide a theoretical generalization bound for CASTLE and demonstrate improved performance against a variety of benchmark methods on a plethora of real and synthetic datasets.

## 2 Related Works

We compare to the related work in the simplest supervised learning setting where we desire learning a function from some features $X$ to a target variable $Y$ given some data of the variables $X$ and $Y$ to improve out-of-sample generalization within the same distribution. This is a significant departure from the branches of machine learning algorithms, such as in semi-supervised learning and domain adaptation, where the regularizer is constructed with information other than variables $X$ and $Y$.

Table 1: Comparison of related works.

| METHOD | FEAT. SEL. | STRUCT. LEARNING | CAUSAL PRED. | TARGET SEL. |
|---|---|---|---|---|
| CAPACITY-BASED | ✓ | ✗ | ✗ | ✗ |
| SAE | ✗ | ✓ | ✗ | ✗ |
| CASTLE | ✓ | ✓ | ✓ | ✓ |

Regularization controls model complexity and mitigates overfitting. $\ell_1$ [29] and $\ell_2$ [30] regularization are commonly used regularization approaches where the former is used when a sparse model is preferred. For deep neural networks, dropout regularization [3, 4, 5] has been shown to be superior in practice to $\ell_p$ regularization techniques. Other capacity-based regularization techniques commonly used in practice include early stopping [31], parameter sharing [31], gradient clipping [32], batch normalization [33], data augmentation [2], weight noise [34], and MixUp [35] to name a few. Norm-based regularizers with sparsity, e.g. Lasso [29], are used to guide feature selection for supervised models. The work of [12] on supervised auto-encoders (SAE) theoretically and empirically shows that adding a reconstruction loss of the input features functions as a regularizer for predictive models. However, this method does not select which features to reconstruct and therefore suffers performance degradation when tasked to reconstruct features that are noise or unrelated to the target variables.

Two existing works [25, 23] attempt to draw the connection between causality and regularization. Based on an analogy between overfitting and confounding in linear models, [25] proposed a method to determine the regularization hyperparameter in linear Ridge or Lasso regression models by estimating the strength of confounding. [23] use causality detectors [36, 27] to weight a sparsity regularizer, e.g. $\ell_1$, for performing non-linear causality analysis and generating multivariate causal hypotheses. Neither of the works has the same objective as us — improving the generalization performance of supervised learning models, nor do they overlap methodologically by using causal DAG discovery.

Causal discovery is an NP-hard problem that requires a brute-force search through a non-convex combinatorial search space, limiting the existing algorithms to reaching global optima for only small problems. Recent approaches have successfully accelerated these methods by using a novel

acyclicity constraint and formulating the causal discovery problem as a continuous optimization over real matrices (avoiding combinatorial search) in the linear [37] and nonlinear [38, 39] cases. CASTLE incorporates these recent causal discovery approaches of [37, 38] to improve regularization for prediction problems in general.

As shown in Table 1, CASTLE regularization provides two additional benefits: causal prediction and target selection. First, CASTLE identifies causal predictors (e.g., causal parents if they exist) rather than correlated features. Furthermore, CASTLE improves upon reconstruction regularization by only reconstructing features that have neighbors in the underlying DAG. We refer to this advantage as "target selection". Collectively these benefits contribute to the improved generalization of CASTLE. Next we introduce our notation (Section 3.1) and provide more details of these benefits (Section 3.2).

## 3    Methodology

In this section, we provide a problem formulation with causal preliminaries for CASTLE. Then we provide a motivational discussion, regularizer methodology, and generalization theory for CASTLE.

### 3.1    Problem Formulation

In the standard supervised learning setting, we denote the input feature variables and target variable, by $\boldsymbol{X} = [X_1, ..., X_d] \in \mathcal{X}$ and $Y \in \mathcal{Y}$, respectively, where $\mathcal{X} \subseteq \mathbb{R}^d$ is a $d$-dimensional feature space and $\mathcal{Y} \subseteq \mathbb{R}$ is a one-dimensional target space. Let $P_{\boldsymbol{X},Y}$ denote the joint distribution of the features and target. Let $[N]$ denote the set $\{1, ..., N\}$. We observe a dataset, $\mathcal{D} = \big\{(\boldsymbol{X}_i, Y_i), i \in [N]\big\}$, consisting of $N$ i.i.d. samples drawn from $P_{\boldsymbol{X},Y}$. The goal of a supervised learning algorithm $\mathcal{A}$ is to find a predictive model, $f_Y : \mathcal{X} \to \mathcal{Y}$, in a hypothesis space $\mathcal{H}$ that can explain the association between the features and the target variable. In the learning algorithm $\mathcal{A}$, the predictive model $\hat{f}_Y$ is trained on a finite number of samples in $\mathcal{D}$, to predict well on the out-of-sample data generated from the same distribution $P_{\boldsymbol{X},Y}$. However, overfitting, a mismatch between training and testing performance of $\hat{f}_Y$, can occur if the hypothesis space $\mathcal{H}$ is too complex and the training data fails to represent the underlying distribution $P_{\boldsymbol{X},Y}$. This motivates the usage of regularization to reduce the hypothesis space's complexity $\mathcal{H}$ so that the learning algorithm $\mathcal{A}$ will only find the desired function to explain the data. Assumptions of the underlying distribution dictate regularization choice. For example, if we believe only a subset of features is associated with the label $Y$, then $\ell_1$ regularization [29] can be beneficial in creating sparsity for feature selection.

CASTLE regularization is based on the assumption that a causal DAG exists among the input features and target variable. In the causal framework of [40], a causal structure of a set of variables $\boldsymbol{X}$ is a DAG in which each vertex $v \in V$ corresponds to a distinct element in $\boldsymbol{X}$, and each edge $e \in E$ represents direct functional relationships between two neighboring variables. Formally, we assume the variables in our dataset satisfy a nonparametric structural equation model (NPSEM) as defined in Definition 1. The word "nonparametric" means we do not make any assumption on the underlying functions $f_i$ in the NPSEM. In this work, we characterize optimal learning by a predictive model as discovering the function $Y = f_Y(\text{Pa}(Y), u_Y)$ in NPSEM [40].

**Definition 1.** *(NPSEMs) Given a DAG $\mathcal{G} = (V = [d+1], E)$, the random variables $\tilde{\boldsymbol{X}} = [Y, \boldsymbol{X}]$ satisfy a NPSEM if*

$$X_i = f_i(Pa(X_i), u_i), \ i \in [d+1],$$

*where $Pa(i)$ is the parents (direct causes) of $X_i$ in $\mathcal{G}$ and $\boldsymbol{u}_{[d+1]}$ are some random noise variables.*

### 3.2    Why CASTLE regularization matters

We now present a graphical example to explain the two benefits of CASTLE mentioned in Section 2, causal prediction and target selection. Consider Figure 1 where we are given nine feature variables $X_1, ..., X_9$ and a target variable $Y$.

**Causal Prediction.** The target variable $Y$ is generated by a function $f_Y(\text{Pa}(Y), u_Y)$ from Definition 1 where the parents of $Y$ are $\text{Pa}(Y) = \{X_2, X_3\}$. In CASTLE regularization, we train a predictive model $\hat{f}_Y$ jointly with learning the DAG among $\boldsymbol{X}$ and $Y$. The features that the model uses to predict $Y$ are the causal parents of $Y$ in the learned DAG. Such a model is sample efficient in uncovering the

true function $f_Y(\text{Pa}(Y), u_Y)$ and generalizes well on the out-of-sample data. Our theoretical analysis in Section 3.4 validates this advantage when there exists a DAG structure among the variables $\boldsymbol{X}$ and $Y$. However, there may exist other variables that predict $Y$ more accurately than the causal parents $\text{Pa}(Y)$. For example, if the function from $Y$ to $X_8$ is a one-to-one linear mapping, we can predict $Y$ trivially from the feature $X_8$. In our objective function introduced later, the prediction loss of $Y$ will be weighted higher than the causal regularizer. Among the predictive models with a similar prediction loss of $Y$, our objective function still prefers to use the model, which minimizes the causal regularizer and uses the causal parents. However, it would favor the easier predictor if one exists and gives a much lower prediction loss of $Y$. In this case, the learned DAG may differ from the true DAG, but we reiterate that we are focused on the problem of generalization rather than causal discovery.

**Target Selection.** Consider the variables $X_5$, $X_6$ and $X_7$ which share parents ($X_2$ and $X_3$) with $Y$ in Figure 1. The functions $X_5 = f_5(X_2, u_5)$, $X_6 = f_6(X_3, u_6)$, and $X_7 = f_7(X_3, u_7)$ may have some learnable similarity (e.g. basis functions and representations) with $Y = f_Y(X_2, X_3, u_Y)$, that we can exploit by training a shared predictive model of $Y$ with the auxiliary task of predicting $X_5$, $X_6$ and $X_7$. From the causal graph topology, CASTLE discovers the optimal features that should act as the auxiliary task for learning $f_Y$. CASTLE learns the related functions jointly in a shared model, which is proven to improve the generalization performance of predicting $Y$ by learning shared basis functions and representations [41].

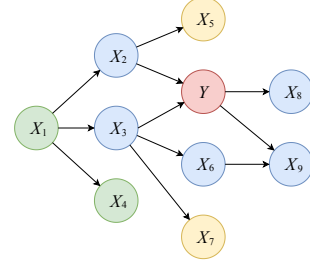

Figure 1: Example DAG.

## 3.3 CASTLE regularization

Let $\tilde{\mathcal{X}} = \mathcal{Y} \times \mathcal{X}$ denote the data space, $P_{(\boldsymbol{X},Y)} = P_{\tilde{\boldsymbol{X}}}$ the data distribution, and $\|\cdot\|_F$ the Frobenius norm. We define random variables $\tilde{\boldsymbol{X}} = [\tilde{X}_1, \tilde{X}_2, ..., \tilde{X}_{d+1}] := [Y, X_1, ..., X_d] \in \tilde{\mathcal{X}}$. Let $\mathbf{X} = [\mathbf{X}_1, ..., \mathbf{X}_d]$ denote the $N \times d$ input data matrix, $\mathbf{Y}$ the $N$-dimensional label vector, $\tilde{\mathbf{X}} = [\mathbf{Y}, \mathbf{X}]$ the $N \times (d+1)$ matrix that contains data of all the variables in the DAG.

To facilitate exposition, we first introduce CASTLE in the linear setting. Here, the parameters are a $(d+1) \times (d+1)$ adjacency matrix $\mathbf{W}$ with zero in the diagonal. The objective function is given as

$$\hat{\mathbf{W}} \in \min_{\mathbf{W}} \frac{1}{N}\|\mathbf{Y} - \tilde{\mathbf{X}}\mathbf{W}_{:,1}\|^2 + \lambda \mathcal{R}_{\text{DAG}}(\tilde{\mathbf{X}}, \mathbf{W}) \tag{1}$$

where $\mathbf{W}_{:,1}$ is the first column of $\mathbf{W}$. We define the DAG regularization loss $\mathcal{R}_{\text{DAG}}(\tilde{\mathbf{X}}, \mathbf{W})$ as

$$\mathcal{R}_{\text{DAG}}(\tilde{\mathbf{X}}, \mathbf{W}) = \mathcal{L}_{\mathbf{W}} + \mathcal{R}_{\mathbf{W}} + \beta \mathcal{V}_{\mathbf{W}}. \tag{2}$$

where $\mathcal{L}_{\mathbf{W}} = \frac{1}{N}\|\tilde{\mathbf{X}} - \tilde{\mathbf{X}}\mathbf{W}\|_F^2$, $R_{\mathbf{W}} = \left(\text{Tr}(e^{\mathbf{W} \odot \mathbf{W}}) - d - 1\right)^2$, $\mathcal{V}_{\mathbf{W}}$ is the $\ell_1$ norm of $\mathbf{W}$, $\odot$ is the Hadamard product, and $e^{\mathbf{M}}$ is the matrix exponential of $\mathbf{M}$. The DAG loss $\mathcal{R}_{\text{DAG}}(\tilde{\mathbf{X}}, \mathbf{W})$ is introduced in [37] for learning linear DAG by continuous optimization. Here we use it as the regularizer for our linear regression model $\mathbf{Y} = \tilde{\mathbf{X}}\mathbf{W}_{:,1} + \boldsymbol{\epsilon}$. From Theorem 1 in [37], we know the graph given by $\mathbf{W}$ is a DAG if and only if $R_{\mathbf{W}} = 0$. The prediction $\hat{\mathbf{Y}} = \tilde{\mathbf{X}}\mathbf{W}_{:,1}$ is the projection of $\mathbf{Y}$ onto the parents of $Y$ in the learned DAG. This increases the stability of linear regression when issues pertaining to collinearity or multicollinearity among the input features appear.

Continuous optimization for learning nonparametric causal DAGs has been proposed in the prior work by [38]. In a similar manner, we also adapt CASTLE to nonlinear cases. Suppose the predictive model for $Y$ and the function generating each feature $X_k$ in the causal DAG are parameterized by an $M$-layer feed-forward neural network $f_\Theta : \tilde{\mathcal{X}} \to \tilde{\mathcal{X}}$ with ReLU activations and layer size $h$. Figure 2 shows the network architecture of $f_\Theta$. This joint network can be instantiated as a $d+1$ sub-network $f_k$ with shared hidden layers, where $f_k$ is responsible for reconstructing the feature $\tilde{X}_k$. We let $\mathbf{W}_1^k$ denote the $h \times (d+1)$ weight matrix in the input layer of $f_k$, $k \in [d+1]$. We set the $k$-th column of $\mathbf{W}_1^k$ to zero such that $f_k$ does not utilize $\tilde{X}_k$ in its prediction of $\tilde{X}_k$. We let $\mathbf{W}_m, m = 2, .., M-1$ denote the weight matrices in the network's shared hidden layers, and $\mathbf{W}_M = [\mathbf{W}_M^1, ..., \mathbf{W}_M^{d+1}]$ denotes the $h \times (d+1)$ weight matrix in the output layer. Explicitly, we define the sub-network $f_k$ as

$$f_k(\tilde{\boldsymbol{X}}) = \phi\big(\cdots\phi\big(\phi(\tilde{\boldsymbol{X}}\mathbf{W}_1^k)\mathbf{W}_2\big)\cdots\mathbf{W}_{M-1}\big)\mathbf{W}_M^k, \tag{3}$$

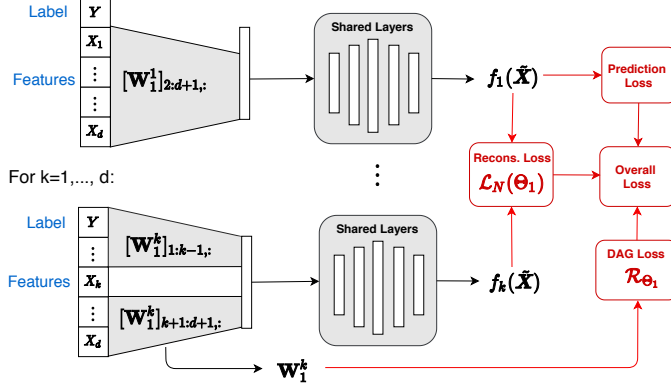

Figure 2: Schematic of CASTLE regularization. Our goal is to have the following tasks: (1) a prediction of a target variable $Y$, and (2) the discovered causal DAG for input features $\boldsymbol{X}$ and $Y$.

where $\phi(\cdot)$ is the ReLU activation function. The function $f_\Theta$ is given as $f_\Theta(\tilde{\boldsymbol{X}}) = [f_1(\tilde{\boldsymbol{X}}), ..., f_{d+1}(\tilde{\boldsymbol{X}})]$. Let $f_\Theta(\tilde{\mathbf{X}})$ denote the prediction for the $N$ samples matrix $\tilde{\mathbf{X}}$ where $[f_\Theta(\tilde{\mathbf{X}})]_{i,k} = f_k(\tilde{\boldsymbol{X}}_i)$, $i \in [N]$ and $k \in [d+1]$. All network parameters are collected into sets as

$$\Theta_1 = \{\mathbf{W}_1^k\}_{k=1}^{d+1}, \ \Theta = \Theta_1 \cup \{\mathbf{W}_m\}_{k=2}^M \tag{4}$$

The training objective function of $f_\Theta$ is

$$\Theta \in \min_\Theta \frac{1}{N} \left\| \mathbf{Y} - [f_\Theta(\tilde{\mathbf{X}})]_{:,1} \right\|^2 + \lambda \mathcal{R}_{\text{DAG}}(\tilde{\mathbf{X}}, f_\Theta). \tag{5}$$

The DAG loss $\mathcal{R}_{\text{DAG}}(\tilde{\mathbf{X}}, f_\Theta)$ is given as

$$\mathcal{R}_{\text{DAG}}(\tilde{\mathbf{X}}, f_\Theta) = \mathcal{L}_N(f_\Theta) + \mathcal{R}_{\Theta_1} + \beta \mathcal{V}_{\Theta_1}. \tag{6}$$

Because the $k$-th column of the input weight matrix $\mathbf{W}_1^k$ is set to zero, $\mathcal{L}_N(f_\Theta) = \frac{1}{N} \left\| \tilde{\mathbf{X}} - f_\Theta(\tilde{\mathbf{X}}) \right\|_F^2$ differs from the standard reconstruction loss in auto-encoders (e.g. SAE) by only allowing the model to reconstruct each feature and target variable from the others. In contrast, auto-encoders reconstruct each feature using all the features including itself. $\mathcal{V}_{\Theta_1}$ is the $\ell_1$ norm of the weight matrices $\mathbf{W}_1^k$ in $\Theta_1$, and the term $\mathcal{R}_{\Theta_1}$ is given as,

$$\mathcal{R}_{\Theta_1} = \left(\text{Tr}\left(e^{\mathbf{M} \odot \mathbf{M}}\right) - d - 1\right)^2, \tag{7}$$

where $\mathbf{M}$ is a $(d+1) \times (d+1)$ matrix such that $[\mathbf{M}]_{k,j}$ is the $\ell_2$-norm of the $k$-th row of the matrix $\mathbf{W}_1^j$. When the acyclicity loss $\mathcal{R}_{\Theta_1}$ is minimized, the sub-networks $f_1, \ldots f_{d+1}$ forms a DAG among the variables; $\mathcal{R}_{\Theta_1}$ obligates the sub-networks to reconstruct only the input features that have neighbors (adjacent nodes) in the learned DAG. We note that converting the nonlinear version of CASTLE into a linear form can be accomplished by removing all the hidden layers and output layers and setting the dimension $h$ of the input weight matrices to be 1 in (3), i.e., $f_k(\tilde{\boldsymbol{X}}) = \tilde{\boldsymbol{X}} \mathbf{W}_1^k$ and $f_\Theta(\tilde{\boldsymbol{X}}) = [\tilde{\boldsymbol{X}} \mathbf{W}_1^1, ..., \tilde{\boldsymbol{X}} \mathbf{W}_1^{d+1}] = \tilde{\boldsymbol{X}} \mathbf{W}$, which is the linear model in (1-2).

**Managing computational complexity.** If the number of features is large, it is computationally expensive to train all the sub-networks simultaneously. We can mitigate this by sub-sampling. At each iteration of gradient descent, we randomly sample a subset of features to reconstruct and only minimize the prediction loss and reconstruction loss on these sub-sampled features. Note that we do not have a hidden confounders issue here, since $Y$ and the sub-sampled features are predicted by all the features except itself. The sparsity DAG constraint on the weight matrices is unchanged at each iteration. In this case, we keep the training complexity per iteration at a manageable level approximately around the computational time and space complexity of training a few networks jointly. We include experiments on CASTLE scalability with respect to input feature size in Appendix C.

### 3.4 Generalization bound for CASTLE regularization

In this section, we analyze theoretically why CASTLE regularization can improve the generalization performance by introducing a generalization bound for our model in Figure 2. Our bound is based

on the PAC-Bayesian learning theory in [42, 43, 44]. Here, we re-interpret the DAG regularizer as a special prior or assumption on the input weight matrices of our model and use existing PAC-Bayes theory to prove the generalization of our algorithm. Traditionally, PAC-Bayes bounds are only applied to randomized models, such as Bayesian or Gibbs classifiers. Here, our bound is applied to our deterministic model by using the recent derandomization formalism from [45, 46]. We acknowledge and note that developing tighter and non-vacuous generalization bounds for deep neural networks is still a challenging and evolving topic in learning theory. The bounds are often stated with many constants from different steps of the proof. For reader convenience, we provide the simplified version of our bound in Theorem 1. The proof, details (e.g., the constants), and discussions about the assumptions are provided in Appendix A. We begin with a few assumptions before stating our bound.

**Assumption 1.** *For any sample $\tilde{X} = (Y, X) \sim P_{\tilde{X}}$, $\tilde{X}$ has bounded $\ell_2$ norm s.t. $\|\tilde{X}\|_2 \leq B$, for some $B > 0$.*

**Assumption 2.** *The loss function $\mathcal{L}(f_\Theta) = \|f_\Theta(\tilde{X}) - \tilde{X}\|^2$ is sub-Gaussian under the distribution $P_{\tilde{X}}$ with a variance factor $s^2$ s.t. $\forall t > 0$, $\mathbb{E}_{P_{\tilde{X}}}\left[\exp\left(t\big(\mathcal{L}(f_\Theta) - \mathcal{L}_P(f_\Theta)\big)\right)\right] \leq \exp(\frac{t^2 s^2}{2})$.*

**Theorem 1.** *Let $f_\Theta : \tilde{\mathcal{X}} \to \tilde{\mathcal{X}}$ be a $M$-layer ReLU feed-forward network with layer size $h$, and each of its weight matrices has the spectral norm bounded by $\kappa$. Then, under Assumptions 1 and 2, for any $\delta, \gamma > 0$, with probability $1 - \delta$ over a training set of $N$ i.i.d samples, for any $\Theta$ in (4), we have:*

$$\mathcal{L}_P(f_\Theta) \leq 4\mathcal{L}_N(f_\Theta) + \frac{1}{N}\left[\mathcal{R}_{\Theta_1} + C_1(\mathcal{V}_{\Theta_1} + \mathcal{V}_{\Theta_2}) + \log\left(\tfrac{8}{\delta}\right)\right] + C_3 \tag{8}$$

*where $\mathcal{L}_P(f_\Theta)$ is the expected reconstruction loss of $\tilde{X}$ under $P_{\tilde{X}}$, $\mathcal{L}_N(f_\Theta)$, $\mathcal{V}_{\Theta_1}$ and $\mathcal{R}_{\Theta_1}$ are defined in (6-7), $\mathcal{V}_{\Theta_2}$ is the $\ell_2$ norm of the network weights in the output and shared hidden layers, and $C_1$ and $C_2$ are some constants depending on $\gamma, d, h, B, s$ and $M$.*

The statistical properties of the reconstruction loss in learning linear DAGs, e.g. $\mathcal{L}_\mathbf{W} = \frac{1}{N}\|\tilde{\mathbf{X}} - \mathbf{W}\tilde{\mathbf{X}}\|_F^2$, have been well studied in the literature: the loss minimizer provably recovers a true DAG with high probability on finite-samples, and hence is consistent for both Gaussian SEM [47] and non-Gaussian SEM [48, 49]. Note also that the regularizer $\mathcal{R}_\mathbf{W}$ or $\mathcal{R}_{\Theta_1}$ are not a part of the results in [47, 48, 49]. However, the works of [37, 38] empirically show that using $\mathcal{R}_\mathbf{W}$ or $\mathcal{R}_{\Theta_1}$ on top of the reconstruction loss leads to more efficient and more accurate DAG learning than existing approaches. Our theoretical result on the reconstruction loss explains the benefit of $\mathcal{R}_\mathbf{W}$ or $\mathcal{R}_{\Theta_1}$ for the generalization performance of predicting $Y$. This provides theoretical support for our CASTLE regularizer in supervised learning. However, the objectives of DAG discovery, e.g., identifying the Markov Blanket of $Y$, is beyond the scope of our analysis.

The bound in (8) justifies $\mathcal{R}_{\Theta_1}$ in general, including linear or nonlinear cases, if the underlying distribution $P_{\tilde{X}}$ is factorized according to some causal DAG. We note that the expected loss $\mathcal{L}_P(f_\Theta)$ is upper bounded by the empirical loss $\mathcal{L}_N(f_\Theta)$, $\mathcal{V}_{\Theta_1}$, $\mathcal{V}_{\Theta_1}$ and $\mathcal{R}_{\Theta_1}$ which measures how close (via acyclicity constraint) the model is to a DAG. From (8) it is obvious that not minimizing $\mathcal{R}_{\Theta_1}$ is an acceptable strategy asymptotically or in the large samples limit (large $N$) because $\mathcal{R}_{\Theta_1}/N$ becomes negligible. This aligns with the consistency theory in [47, 48, 49] for linear models. However for small $N$, a preferred strategy is to train a model $f_\Theta$ by minimizing $\mathcal{L}_N(f_\Theta)$ and $\mathcal{R}_{\Theta_1}$ jointly. This would be trivial because the samples are generated under the DAG structure in $P_{\tilde{X}}$. Minimizing $\mathcal{R}_{\Theta_1}$ can decrease the upper bound of $\mathcal{L}_P(f_\Theta)$ in (8), improve the generalization performance of $f_\Theta$, as well as facilitate the convergence of $f_\Theta$ to the true model.

If $P_{\tilde{X}}$ does not correspond to any causal DAG, such as image data, then there will be a trade-off between minimizing $\mathcal{R}_{\Theta_1}$ and $\mathcal{L}_N(f_\Theta)$. In this case, $\mathcal{R}_{\Theta_1}$ becomes harder to minimize, and generalization may not benefit from adding CASTLE. However, this is a rare case since causal structure exists in most datasets inherently. Our experiments demonstrate that CASTLE regularization outperforms popular regularizers on a variety of datasets in the next section.

## 4 Experiments

In this section, we empirically evaluate CASTLE as a regularization method for improving generalization performance. We present our benchmark methods and training architecture, followed by our synthetic and publicly available data results.

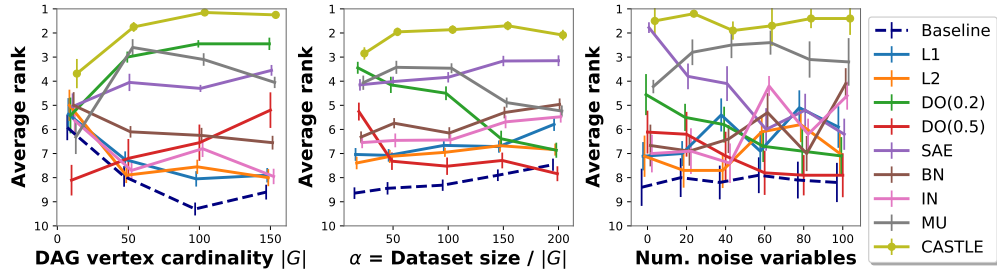

Figure 3: Experiments on synthetic data. The $y$-axis is the average rank ($\pm$ standard deviation) of each regularizer on the test set over each synthetic DAG. We show the average rank as we increase the number of features or vertex cardinality $|G|$ (**left**), increase the dataset size normalized by the vertex cardinality $|G|$ (**center**), and as we increase the number of noise (neighborless) variables (**right**).

**Benchmarks.** We benchmark CASTLE against common regularizers that include: early stopping (Baseline) [31], L1 [29], L2 [30], dropout [3] with drop rate of 20% and 50% denoted as DO(0.2) and DO(0.5) respectively, SAE [12], batch normalization (BN) [33], data augmentation or input noise (IN) [2], and MixUp (MU) [35], in no particular order. For each regularizer with tunable hyperparameters we performed a standard grid search. For the weight decay regularizers L1 and L2 we searched for $\lambda_{\ell_p} \in \{0.1, 0.01, 0.001\}$, and for input noise we use a Gaussian noise with mean of 0 and standard deviation $\sigma \in \{0.1, 0.01, 0.01\}$. L1 and L2 were applied at every dense layer. BN and DO were applied after every dense layer and active only during training. Because each regularization method converges at different rates, we use early stopping on a validation set to terminate each benchmark training, which we refer to as our Baseline.

**Network architecture and training.** We implemented CASTLE in `Tensorflow`[2]. Our proposed architecture is comprised of $d + 1$ sub-networks with shared hidden layers, as shown in Figure 2. In the linear case, $\mathcal{V}_{\mathbf{W}}$ is the $\ell_1$ norm of $\mathbf{W}$. In the nonlinear case, $\mathcal{V}_{\Theta_1}$ is the $\ell_1$ norm of the input weight matrices $\mathbf{W}_1^k, k \in [d+1]$. To make a clear comparison with L2 regularization, we exclude the capacity term $\mathcal{V}_{\Theta_2}$ from CASTLE, although it is a part of our generalization bound in (8). Since we predict the target variable as our primary task, we benchmark CASTLE against this common network architecture. Specifically, we use a network with two hidden layers of $d + 1$ neurons with ReLU activation. Each benchmark method is initialized and seeded identically with the same random weights. For dataset preprocessing, all continuous variables are standardized with a mean of 0 and a variance of 1. Each model is trained using the Adam optimizer with a learning rate of 0.001 for up to a maximum of 200 epochs. An early stopping regime halts training with a patience of 30 epochs.

## 4.1 Regularization on Synthetic Data

**Synthetic data generation.** Given a DAG $G$, we generate functional relationships between each variable and its respective parent(s) with additive Gaussian noise applied to each variable with a mean of 0 and variance of 1. In the linear case, each variable is equal to the sum of its parents plus noise. For the nonlinear case, each variable is equal to the sum of the sigmoid of its parents plus noise. We provide further details on our synthetic DGP and pseudocode in Appendix B. Consider Table 2, using our nonlinear DGP we generated 1000 test samples according to the DAG in Figure 1. We then used 10-fold cross-validation to train and validate each benchmark on varying training sets of size $n$.

Table 2: Experiments on nonlinear synthetic data of size $n$ generated according to Fig. 1 in terms of MSE ($\pm$ standard deviation)

| Regularizer | $n = 500$ | $n = 1000$ | $n = 5000$ |
|---|---|---|---|
| Baseline | $0.83 \pm 0.03$ | $0.80 \pm 0.04$ | $0.73 \pm 0.02$ |
| L1 | $0.81 \pm 0.05$ | $0.79 \pm 0.03$ | $0.71 \pm 0.02$ |
| L2 | $0.81 \pm 0.05$ | $0.77 \pm 0.02$ | $0.71 \pm 0.01$ |
| DO(0.2) | $0.80 \pm 0.04$ | $0.79 \pm 0.01$ | $0.70 \pm 0.02$ |
| DO(0.5) | $0.79 \pm 0.02$ | $0.78 \pm 0.04$ | $0.70 \pm 0.02$ |
| SAE | $0.79 \pm 0.03$ | $0.77 \pm 0.04$ | $0.69 \pm 0.02$ |
| BN | $0.81 \pm 0.04$ | $0.79 \pm 0.03$ | $0.72 \pm 0.02$ |
| IN | $0.82 \pm 0.05$ | $0.78 \pm 0.04$ | $0.71 \pm 0.02$ |
| MU | $0.79 \pm 0.05$ | $0.78 \pm 0.04$ | $0.72 \pm 0.08$ |
| CASTLE | $\mathbf{0.77 \pm 0.02}$ | $\mathbf{0.75 \pm 0.04}$ | $\mathbf{0.68 \pm 0.02}$ |

Each model was evaluated on the test set from weights saved at the lowest validation error. Table 2 shows that CASTLE improves over all experimental benchmarks. We present similar results for our linear experiments in Appendix B.

Table 3: Comparison of benchmark regularizers on regression and classification in terms of test MSE and AUROC (± standard deviation), respectively, for experiments on real datasets using 10-fold cross-validation. Bold denotes best performance. For conciseness we show only a subset of the benchmarks. The full version of this table is in Appendix C along with results on additional datasets.

| Dataset | Baseline | L1 | Dropout 0.2 | SAE | Batch Norm | Input Noise | MixUp | CASTLE |
|---|---|---|---|---|---|---|---|---|
| | | | | Regression (MSE) | | | | |
| BH | $0.141 \pm 0.023$ | $0.137 \pm 0.025$ | $0.168 \pm 0.032$ | $0.148 \pm 0.027$ | $0.139 \pm 0.021$ | $0.137 \pm 0.018$ | $0.194 \pm 0.064$ | $\mathbf{0.123 \pm 0.016}$ |
| WQ | $0.747 \pm 0.038$ | $0.747 \pm 0.043$ | $0.738 \pm 0.029$ | $0.727 \pm 0.030$ | $0.723 \pm 0.039$ | $0.771 \pm 0.036$ | $0.712 \pm 0.018$ | $\mathbf{0.708 \pm 0.030}$ |
| FB | $0.758 \pm 1.017$ | $0.663 \pm 0.796$ | $0.429 \pm 0.449$ | $0.372 \pm 0.168$ | $0.705 \pm 0.396$ | $0.609 \pm 0.511$ | $0.385 \pm 0.208$ | $\mathbf{0.246 \pm 0.153}$ |
| BC | $0.359 \pm 0.061$ | $0.342 \pm 0.037$ | $0.334 \pm 0.030$ | $0.322 \pm 0.021$ | $0.325 \pm 0.024$ | $0.319 \pm 0.022$ | $0.322 \pm 0.030$ | $\mathbf{0.318 \pm 0.036}$ |
| SP | $0.416 \pm 0.108$ | $0.421 \pm 0.181$ | $0.285 \pm 0.042$ | $0.228 \pm 0.022$ | $0.318 \pm 0.062$ | $0.389 \pm 0.095$ | $0.267 \pm 0.072$ | $\mathbf{0.200 \pm 0.020}$ |
| CM | $0.536 \pm 0.103$ | $0.574 \pm 0.125$ | $0.327 \pm 0.025$ | $0.387 \pm 0.034$ | $0.470 \pm 0.047$ | $0.495 \pm 0.081$ | $0.376 \pm 0.030$ | $\mathbf{0.326 \pm 0.031}$ |
| | | | | Classification (AUROC) | | | | |
| CC | $0.764 \pm 0.009$ | $0.766 \pm 0.007$ | $0.776 \pm 0.009$ | $0.774 \pm 0.012$ | $0.773 \pm 0.009$ | $0.772 \pm 0.012$ | $0.778 \pm 0.009$ | $\mathbf{0.787 \pm 0.007}$ |
| PD | $0.799 \pm 0.008$ | $0.793 \pm 0.013$ | $0.797 \pm 0.010$ | $0.796 \pm 0.010$ | $0.773 \pm 0.024$ | $0.796 \pm 0.013$ | $0.802 \pm 0.016$ | $\mathbf{0.817 \pm 0.004}$ |
| BC | $0.721 \pm 0.018$ | $0.726 \pm 0.011$ | $0.718 \pm 0.024$ | $0.605 \pm 0.068$ | $0.727 \pm 0.012$ | $0.722 \pm 0.026$ | $0.700 \pm 0.055$ | $\mathbf{0.731 \pm 0.010}$ |
| LV | $0.559 \pm 0.061$ | $0.594 \pm 0.020$ | $0.579 \pm 0.053$ | $0.542 \pm 0.095$ | $0.583 \pm 0.026$ | $0.597 \pm 0.041$ | $0.553 \pm 0.092$ | $\mathbf{0.595 \pm 0.032}$ |
| SH | $0.915 \pm 0.015$ | $0.921 \pm 0.006$ | $0.922 \pm 0.017$ | $0.701 \pm 0.205$ | $0.913 \pm 0.013$ | $0.922 \pm 0.005$ | $0.921 \pm 0.005$ | $\mathbf{0.929 \pm 0.007}$ |
| RP | $0.782 \pm 0.071$ | $0.801 \pm 0.013$ | $0.743 \pm 0.052$ | $0.774 \pm 0.103$ | $0.802 \pm 0.018$ | $0.796 \pm 0.009$ | $0.730 \pm 0.043$ | $\mathbf{0.814 \pm 0.014}$ |

**Dissecting CASTLE.** In the synthetic environment, we know the causal relationships with certainty. We analyze three aspects of CASTLE regularization using synthetic data. Because we are comparing across randomly simulated DAGs with differing functional relationships, the magnitude of regression testing error will vary between runs. We examine the model performance in terms of each model's average rank over each fold to normalize this. If we have $r$ regularizers, the best and worst possible rank is one and $r$, respectively (i.e., the higher the rank the better). We used 10-fold cross-validation to terminate model training and tested each model on a held-out test set of 1000 samples.

First, we examine the impact of increasing the feature size or DAG vertex cardinality $|G|$. We do this by randomly generating a DAG of size $|G| \in \{10, 50, 100, 150\}$ with $50|G|$ training samples. We repeat this ten times for each DAG cardinality. On the left-hand side of Fig. 3, CASTLE has the highest rank of all benchmarks and does not degrade with increasing $|G|$. Second, we analyze the impact of increasing dataset size. We randomly generate DAGs of size $|G| \in \{10, 50, 100, 150\}$, which we use to create datasets of $\alpha|G|$ samples, where $\alpha \in \{20, 50, 100, 150, 200\}$. We repeat this ten times for each dataset size. In the middle plot of Figure 3, we see that CASTLE has superior performance for all dataset sizes, and as expected, all benchmark methods (except for SAE) start to converge about the average rank at large data sizes ($\alpha = 200$). Third, we analyze our method's sensitivity to noise variables, i.e., variables disconnected to the target variable in $G$. We randomly generate DAGs of size $|G| = 50$ to create datasets with $50|G|$ samples. We randomly add $v \in \{20i\}_{i=0}^{5}$ noise variables normally distributed with 0 mean and unit variance. We repeat this process for ten different DAG instantiations. The results on the right-hand side of Figure 3 show that our method is not sensitive to the existence of disconnected noise variables, whereas SAE performance degrades with the increase of uncorrelated input features. This highlights the benefit of target selection based on the DAG topology. In Appendix C, we provide an analysis of adjacency matrix weights that are learned under various random DAG configurations, e.g., target with parents, orphaned target, etc. There, we highlight CASTLE in comparison to SAE for target selection by showing that the adjacency matrix weights for noise variables are near zero. We also provide a sensitivity analysis on the parameter $\lambda$ from (5) and results for additional experiments demonstrating that CASTLE does not reconstruct noisy (neighborless) variables in the underlying causal DAG.

## 4.2 Regularization on Real Data

We perform regression and classification experiments on a spectrum of publicly available datasets from [50] including Boston Housing (BH), Wine Quality (WQ), Facebook Metrics (FB), Bioconcentration (BC), Student Performance (SP), Community (CM), Contraception Choice (CC), Pima Diabetes (PD), Las Vegas Ratings (LV), Statlog Heart (SH), and Retinopathy (RP). For each dataset, we randomly reserve 20% of the samples for a testing set. We perform 10-fold cross-validation on the remaining 80%. As the results show in Table 3, CASTLE provides improved regularization across all datasets for both regression and classification tasks. Additionally, CASTLE consistently ranks as the top regularizer (graphically shown in Appendix C.3), with no definitive benchmark

method coming in as a consensus runner-up. This emphasizes the stability of CASTLE as a reliable regularizer. In Appendix C, we provide additional experiments on several other datasets, an ablation study highlighting our sources of gain, and real-world dataset statistics.

## 5    Conclusion

We have introduced CASTLE regularization, a novel regularization method that jointly learns the causal graph to improve generalization performance in comparison to existing capacity-based and reconstruction-based regularization methods. We used existing PAC-Bayes theory to provide a theoretical generalization bound for CASTLE. We have shown experimentally that CASTLE is insensitive to increasing feature dimensionality, dataset size, and uncorrelated noise variables. Furthermore, we have shown that CASTLE regularization improves performance on a plethora of real datasets and, in the worst case, never degrades performance. We hope that CASTLE will play a role as a general-purpose regularizer that can be leveraged by the entire machine learning community.

## Broader Impact

One of the big challenges of machine learning, and deep learning in particular, is generalization to out-of-sample data. Regularization is necessary and used to prevent overfitting thereby promoting generalization. In this work, we have presented a novel regularization method inspired by causality. Since the applicability of our approach spans all problems where causal relationships exist between variables, there are countless beneficiaries of our research. Apart from the general machine learning community, the beneficiaries of our research include practitioners in the social sciences (sociology, psychology, etc.), natural sciences (physics, biology, etc.), and healthcare among countless others. These fields have already been exploiting causality for some time and serve as a natural launch-pad for deploying and leveraging CASTLE. With that said, our method does not immediately apply to certain architectures, such as CNNs, where causal relationships are ambiguous or perhaps non-existent.

## Acknowledgments

This work was supported by GlaxoSmithKline (GSK), the US Office of Naval Research (ONR), and the National Science Foundation (NSF) 1722516. We thank all reviewers for their invaluable comments and suggestions.

## Footnotes

[2]Code is provided at `https://bitbucket.org/mvdschaar/mlforhealthlabpub`.

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
