[Supplementary Material]

# A Proof of Theorem 1

In this paper, we consider learning a causal DAG as our regularizer. We use a squared loss in our objective function. We find the sub-Gaussian assumption is more realistic than the bounded assumption because the squared loss function is usually unbounded but with strong tail decay property. We also assume our network weights have bounded spectral norm. Since a generalization bound considers all the models in the hypothesis class, the hypothesis class's capability should be restricted to upper bound the expected loss for all the regression models in the class. However, this assumption is not necessary for classification problems, since it is possible to normalize the network weight. The network still has the same prediction as before normalization due to the ReLU homogeneity.

**Theorem 1.** *Let $f_\Theta : \tilde{\mathcal{X}} \to \tilde{\mathcal{X}}$ be a $M$-layer ReLU feed-forward network with layer size $h$, and each of its weight matrices has the spectral norm bounded by $\kappa$. Then, under Assumptions 1 and 2, for any $\delta, \gamma > 0$, with probability $1 - \delta$ over a training set of $N$ i.i.d samples, for any $\Theta$ in (4), we have:*

$$\mathcal{L}_P(f_\Theta) \leq 4\mathcal{L}_N(f_\Theta) + \frac{1}{N}\Big[\mathcal{R}_{\Theta_1} + C_1(\mathcal{V}_{\Theta_1} + \mathcal{V}_{\Theta_2}) + \log\big(\tfrac{8}{\delta}\big)\Big] + C_2 \tag{9}$$

*where $\mathcal{L}_P(f_\Theta)$ is the expected reconstruction loss of $\tilde{\boldsymbol{X}}$ under $P_{\tilde{\boldsymbol{X}}}$, $\mathcal{L}_N(f_\Theta)$, $\mathcal{V}_{\Theta_1}$ and $\mathcal{R}_{\Theta_1}$ are defined in (6-7), $\mathcal{V}_{\Theta_2}$ is the $\ell_2$ norm of the network weights in the output and shared hidden layers, $C_1$ and $C_2$ are given as $C_1 = (\gamma^{-1}\zeta(d+1)\kappa^{M-1})^2$, $C_2 = s^2 + 6\gamma$, and $\zeta = MBe\big[2\log(2eh)\big]^{1/2}$.*

*Proof.* Our proof consists of three steps: (1) We convert the existing PAC-Bayes bound for a randomized model $f_{\Theta_{\boldsymbol{u}}}$ to a deterministic model $f_\Theta$; (2) We upper bound the KL divergence in the PAC-Bayes bound by the capability terms (i.e. the regularizers) of our model; (3) We discuss how to choose the constants in our bound to make our result universal.

**Step 1.** We let $\tilde{\Theta}_{\boldsymbol{u}}$ denote the $\Theta$ in which we perturb each parameter by a random perturbation $u$ drawn from some Gaussian distribution. We collect all the random perturbation into one vector $\boldsymbol{u}$, and $\boldsymbol{u} \sim N(\boldsymbol{0}, \sigma^2 I)$. We let $Q_{\Theta_{\boldsymbol{u}}}$ denote the distribution of $\Theta_{\boldsymbol{u}}$, and $P_{\Theta_{\boldsymbol{u}}}$ denote our prior on $\Theta_{\boldsymbol{u}}$. For $\mathcal{L}_N(f_{\Theta_{\boldsymbol{u}}})$, we have

$$
\begin{aligned}
\mathbb{E}_{\boldsymbol{u}}\big[\mathcal{L}_N(f_{\Theta_{\boldsymbol{u}}})\big] &= \mathbb{E}_{\boldsymbol{u}}\left[\frac{1}{N}\sum_{i=1}^{N}\big\|f_{\Theta_{\boldsymbol{u}}}(\tilde{\boldsymbol{X}}_i) - f_\Theta(\tilde{\boldsymbol{X}}_i) + f_\Theta(\tilde{\boldsymbol{X}}_i) - \tilde{\boldsymbol{X}}_i\big\|^2\right] \\
&= \mathbb{E}_{\boldsymbol{u}}\left[\frac{1}{N}\sum_{i=1}^{N}\big\|f_{\Theta_{\boldsymbol{u}}}(\tilde{\boldsymbol{X}}_i) - f_\Theta(\tilde{\boldsymbol{X}}_i)\big\|^2\right] + \frac{1}{N}\sum_{i=1}^{N}\big\|f_\Theta(\tilde{\boldsymbol{X}}_i) - \tilde{\boldsymbol{X}}_i\big\|^2 \\
&\quad + \mathbb{E}_{\boldsymbol{u}}\left[\frac{2}{N}\sum_{i=1}^{N}\big(f_{\Theta_{\boldsymbol{u}}}(\tilde{\boldsymbol{X}}_i) - f_\Theta(\tilde{\boldsymbol{X}}_i)\big)\big(f_\Theta(\tilde{\boldsymbol{X}}_i) - \tilde{\boldsymbol{X}}_i\big)\right] \\
&\leq \mathbb{E}_{\boldsymbol{u}}\left[\frac{1}{N}\sum_{i=1}^{N}\big\|f_{\Theta_{\boldsymbol{u}}}(\tilde{\boldsymbol{X}}_i) - f_\Theta(\tilde{\boldsymbol{X}}_i)\big\|^2\right] + \frac{1}{N}\sum_{i=1}^{N}\big\|f_\Theta(\tilde{\boldsymbol{X}}_i) - \tilde{\boldsymbol{X}}_i\big\|^2 \\
&\quad + \mathbb{E}_{\boldsymbol{u}}\left[\frac{1}{N}\sum_{i=1}^{N}\big\|f_{\Theta_{\boldsymbol{u}}}(\tilde{\boldsymbol{X}}_i) - f_\Theta(\tilde{\boldsymbol{X}}_i)\big\|^2\right] + \frac{1}{N}\sum_{i=1}^{N}\big\|f_\Theta(\tilde{\boldsymbol{X}}_i) - \tilde{\boldsymbol{X}}_i\big\|^2 \\
&\leq 2\gamma + 2\mathcal{L}_N(f_\Theta)
\end{aligned}
\tag{10}
$$

Similarly, we have

$$
\begin{aligned}
\mathcal{L}_P(f_\Theta) &= \mathbb{E}_P\mathbb{E}_{\boldsymbol{u}}\left[\big\|f_\Theta(\tilde{\boldsymbol{X}}) - f_{\Theta_{\boldsymbol{u}}}(\tilde{\boldsymbol{X}}) + f_{\Theta_{\boldsymbol{u}}}(\tilde{\boldsymbol{X}}) - \tilde{\boldsymbol{X}}\big\|^2\right] \\
&\leq 2\gamma + 2\mathbb{E}_{\boldsymbol{u}}\big[L_P(f_{\Theta_{\boldsymbol{u}}})\big]
\end{aligned}
\tag{11}
$$

where we let $\gamma$ be a constant such that $\max_{\tilde{\boldsymbol{X}} \in \mathcal{X}} \mathbb{E}_{\boldsymbol{u}}\big[\|f_{\Theta_{\boldsymbol{u}}}(\tilde{\boldsymbol{X}}) - f_\Theta(\tilde{\boldsymbol{X}})\|^2\big] \leq \gamma$. It is the upper bound for the maximum expected change of the network output when the weights are perturbed, thereby the network's sharpness as defined in [51].

Using the Corollary 4 in [52] and Lemma 1 in [45], we have the following PAC Bayes bound for the randomized model $f_{\Theta_{\boldsymbol{u}}}$. Given a prior distribution $P_{\Theta_{\boldsymbol{u}}}$ over the set of predictors that is independent

of the training data, the PAC-Bayes theorem states that with probability at least $1 - \delta$, over $N$ i.i.d training samples, the expected error of $f_{\Theta_{\boldsymbol{u}}}$ can be bounded as follows,

$$\mathbb{E}_{\boldsymbol{u}}\big[\mathcal{L}_P(f_{\Theta_{\boldsymbol{u}}})\big] \leq \mathbb{E}_{\boldsymbol{u}}\big[\mathcal{L}_N(f_{\Theta_{\boldsymbol{u}}})\big] + \tfrac{1}{N}\Big[2\operatorname{KL}(Q_{\Theta_{\boldsymbol{u}}}\|P_{\Theta_{\boldsymbol{u}}}) + \log(\tfrac{8}{\delta})\Big] + \tfrac{1}{2}s^2 \qquad (12)$$

If we upper bound $\mathbb{E}_{\boldsymbol{u}}\big[\mathcal{L}_P(f_{\Theta_{\boldsymbol{u}}})\big]$ in (11) by (12), we have

$$
\begin{aligned}
\mathcal{L}_P(f_\Theta) &\leq 2\gamma + 2\mathbb{E}_{\boldsymbol{u}}\big[\mathcal{L}_N(f_{\Theta_{\boldsymbol{u}}})\big] + \tfrac{2}{N}\Big[2\operatorname{KL}(Q_{\Theta_{\boldsymbol{u}}}\|P_{\Theta_{\boldsymbol{u}}}) + \log(\tfrac{8}{\delta})\Big] + s^2 \\
&\leq 4\mathcal{L}_N(f_\Theta) + \tfrac{2}{N}\Big[2\operatorname{KL}(Q_{\Theta_{\boldsymbol{u}}}\|P_{\Theta_{\boldsymbol{u}}}) + \log(\tfrac{8}{\delta})\Big] + C_2
\end{aligned}
\qquad (13)
$$

where the last inequality is achieved by (10), and $C_2 = s^2 + 6\gamma$.

**Step 2**. For convenience, we restate the parameter set $\Theta$ in (4) here,

$$\Theta_1 = \{\mathbf{W}_1^k\}_{k=1}^{d+1}, \ \Theta = \Theta_1 \cup \{\mathbf{W}_m\}_{k=2}^M$$

Now we write the distribution $Q_{\Theta_{\boldsymbol{u}}}$ and $P_{\Theta_{\boldsymbol{u}}}$ explicitly. Without loss of generality, we assume $Q_{\Theta_{\boldsymbol{u}}}$ and $P_{\Theta_{\boldsymbol{u}}}$ have the same standard deviation $\sigma^2$. First, $Q_{\Theta_{\boldsymbol{u}}}$ is given as $Q_{\Theta_{\boldsymbol{u}}} = Q_{\Theta_{\boldsymbol{u}}}^{(1)} Q_{\Theta_{\boldsymbol{u}}}^{(2)}$, where $Q_{\Theta_{\boldsymbol{u}}}^{(1)} = N(z_{\Theta_{\boldsymbol{u}},1}; z_{\Theta_1}, 1)$, and

$$Q_{\Theta_{\boldsymbol{u}}}^{(2)} = \prod_{k=1}^{d+1} N(\mathbf{W}_{\boldsymbol{u},1}^k; \mathbf{W}_1^k, \sigma^2 I) \prod_{m=2}^{M} N(\mathbf{W}_{\boldsymbol{u},m}; \mathbf{W}_m, \sigma^2 I).$$

And $P_\Theta$ is given as $P_{\Theta_{\boldsymbol{u}}} = P_{\Theta_{\boldsymbol{u}}}^{(1)} P_{\Theta_{\boldsymbol{u}}}^{(2)}$, where $P_{\Theta_{\boldsymbol{u}}}^{(1)} = N(z_{\Theta_{\boldsymbol{u}},1}; d+1, 1)$, and

$$P_{\Theta_{\boldsymbol{u}}}^{(2)} = \prod_{k=1}^{d+1} N(\mathbf{W}_{\boldsymbol{u},1}^k; \mathbf{0}, \sigma^2 I) \prod_{m=2}^{M} N(\mathbf{W}_{\boldsymbol{u},m}; \mathbf{0}, \sigma^2 I).$$

The variable $z_{\Theta_{\boldsymbol{u}},1}$ is given as,

$$z_{\Theta_{\boldsymbol{u}},1} = \operatorname{Tr}\big(e^{\mathbf{M}_{\boldsymbol{u}} \odot \mathbf{M}_{\boldsymbol{u}}}\big)$$

where $\mathbf{M}_{\boldsymbol{u}}$ is a $(d+1) \times (d+1)$ matrix such that $[\mathbf{M}_{\boldsymbol{u}}]_{k,j}$ is the $\ell_2$-norm of the $k$-th row of the matrix $\mathbf{W}_{\boldsymbol{u},1}^j$. The variable $z_{\Theta_1}$ is defined in the same way as $z_{\Theta_{\boldsymbol{u}},1}$ but on the parameters without perturbations. Here, we use Gaussian distributions for $z$'s for simplicity in our deterministic model. Formally, in Bayesian inference, we may consider using truncated normal or exponential priors for $z$'s since we know $z_{\Theta_{\boldsymbol{u}},1} = \operatorname{Tr}(I) + \operatorname{Tr}(\mathbf{M}_{\boldsymbol{u}} \odot \mathbf{M}_{\boldsymbol{u}}) + \cdots \geq d+1$ using the power series of matrix exponential and the fact that each element of $\mathbf{M}_{\boldsymbol{u}}$ is non-negative. Now we upper bound the KL divergence as follows,

$$
\begin{aligned}
\operatorname{KL}(Q_{\Theta_{\boldsymbol{u}}}\|P_{\Theta_{\boldsymbol{u}}}) &= \int Q_{\Theta_{\boldsymbol{u}}}^{(1)} Q_{\Theta_{\boldsymbol{u}}}^{(2)} \log\Big(\tfrac{Q_{\Theta_{\boldsymbol{u}}}^{(1)} Q_{\Theta_{\boldsymbol{u}}}^{(2)}}{P_{\Theta_{\boldsymbol{u}}}^{(1)} P_{\Theta_{\boldsymbol{u}}}^{(2)}}\Big) \, d\Theta_{\boldsymbol{u}} \\
&= \int Q_{\Theta_{\boldsymbol{u}}}^{(1)} Q_{\Theta_{\boldsymbol{u}}}^{(2)} \log\Big(\tfrac{Q_{\Theta_{\boldsymbol{u}}}^{(1)}}{P_{\Theta_{\boldsymbol{u}}}^{(1)}}\Big) \, d\Theta_{\boldsymbol{u}} + \int Q_{\Theta_{\boldsymbol{u}}}^{(1)} Q_{\Theta_{\boldsymbol{u}}}^{(2)} \log\Big(\tfrac{Q_{\Theta_{\boldsymbol{u}}}^{(2)}}{P_{\Theta_{\boldsymbol{u}}}^{(2)}}\Big) \, d\Theta_{\boldsymbol{u}} \\
&\leq \int Q_{\Theta_{\boldsymbol{u}}}^{(1)} \log\Big(\tfrac{Q_{\Theta_{\boldsymbol{u}}}^{(1)}}{P_{\Theta_{\boldsymbol{u}}}^{(1)}}\Big) \, d\Theta_{\boldsymbol{u}} + \int Q_{\Theta_{\boldsymbol{u}}}^{(2)} \log\Big(\tfrac{Q_{\Theta_{\boldsymbol{u}}}^{(2)}}{P_{\Theta_{\boldsymbol{u}}}^{(2)}}\Big) \, d\Theta_{\boldsymbol{u}} \qquad (14) \\
&= \tfrac{1}{2}\big[z_{\Theta_1} - (d+1)\big]^2 + \tfrac{1}{2\sigma^2}\Big(\sum_{k=1}^{d+1} \|\mathbf{W}_1^k\|_F^2 + \sum_{m=2}^{M} \|\mathbf{W}_m\|_F^2\Big) \\
&\leq \tfrac{1}{2}\mathcal{R}_{\theta_1} + \tfrac{1}{2\sigma^2}(\mathcal{V}_{\Theta_1} + \mathcal{V}_{\Theta_2})
\end{aligned}
$$

where the last inequality is achieved using the fact that the Euclidean norm of any vector is bounded by its $\ell_1$-norm. Let $C_1 = \tfrac{1}{\sigma^2}$. Bounding the KL divergence in (13) with (14) gives that

$$\mathcal{L}_P(f_\Theta) \leq 4\mathcal{L}_N(f_\Theta) + \tfrac{2}{N}\Big[\mathcal{R}_{\theta_1} + C_1(\mathcal{V}_{\Theta_1} + \mathcal{V}_{\Theta_2}) + \log(\tfrac{8}{\delta})\Big] + C_2 \qquad (15)$$

**Step 3.** Recall that $\gamma$ is the upper bound for $\max_{\tilde{\boldsymbol{X}} \in \mathcal{X}} \mathbb{E}_{\boldsymbol{u}}\big[\|f_{\Theta_{\boldsymbol{u}}}(\tilde{\boldsymbol{X}}) - f_\Theta(\tilde{\boldsymbol{X}})\|^2\big]$, the expected maximum change of the network output when the weights are perturbed by $\boldsymbol{u} \sim N(\mathbf{0}, \sigma^2 I)$. We

now derive the constant $\gamma$ based on $\sigma^2$, the input upper bound $B$ in Assumption 1. Our network uses ReLU activation functions in the hidden layers. The ReLU function $\phi(\cdot)$ is 1-Lipschitz. This proof is similar to Lemma 2 in [45]. Let $\|\cdot\|_2$ denote the spectral norm. We define $\Delta_k^{M-1}$ as the output difference in the last hidden layer:

$$
\begin{aligned}
\Delta_k^{M-1} = \big\|\phi\big(\cdots\phi\big(\phi(\tilde{\boldsymbol{X}}[\mathbf{W}_1^k + \mathbf{U}_1^k])[\mathbf{W}_2 + \mathbf{U}_2]\big)\cdots[\mathbf{W}_{M-1} + \mathbf{U}_{M-1}]\big) \\
- \phi\big(\cdots\phi\big(\phi(\tilde{\boldsymbol{X}}\mathbf{W}_1^k)\mathbf{W}_2\big)\cdots\mathbf{W}_{M-1}\big)\big\|
\end{aligned}
$$

We have

$$
\begin{aligned}
\Delta_k^M &= \big([f_\Theta(\tilde{\boldsymbol{X}})]_k - [f_{\Theta_{\boldsymbol{u}}}(\tilde{\boldsymbol{X}})]_k\big)^2 \\
&= \Delta_k^{M-1}\big[\|\mathbf{W}_M\|_2 + \|\mathbf{U}_M\|_2\big] + \|\tilde{\boldsymbol{X}}\|\|\mathbf{U}_M\|_2\|\mathbf{W}_1^k\|_2\prod_{m=2}^{M-1}\|\mathbf{W}_m\|_2 \\
&\le (1+\tfrac{1}{M})\|\mathbf{W}_M\|_2\Delta_k^{M-1} + \tfrac{\|\mathbf{U}_M\|_2}{\|\mathbf{W}_M\|_2}\|\tilde{\boldsymbol{X}}\|\|\mathbf{W}_1^k\|_2\prod_{m=2}^{M}\|\mathbf{W}_m\|_2 \\
&\le (1+\tfrac{1}{M})\|\mathbf{W}_M\|_2\Big((1+\tfrac{1}{M})\|\mathbf{W}_{M-1}\|_2\Delta_k^{M-2} + \tfrac{\|\mathbf{U}_{M-1}\|_2}{\|\mathbf{W}_{M-1}\|_2}\|\tilde{\boldsymbol{X}}\|\|\mathbf{W}_1^k\|_2\prod_{m=2}^{M-1}\|\mathbf{W}_m\|_2\Big) \\
&\quad + \tfrac{\|\mathbf{U}_M\|_2}{\|\mathbf{W}_M\|_2}\|\tilde{\boldsymbol{X}}\|\|\mathbf{W}_1^k\|_2\prod_{m=2}^{M}\|\mathbf{W}_m\|_2 \\
&\le (1+\tfrac{1}{M})^2\Delta_k^{M-2}\prod_{m=M-1}^{M}\|\mathbf{W}_m\|_2 + \sum_{m=0}^{1}(1+\tfrac{1}{M})^m\tfrac{\|\mathbf{U}_{M-m}\|_2}{\|\mathbf{W}_{M-m}\|_2}\|\tilde{\boldsymbol{X}}\|\|\mathbf{W}_1^k\|_2\prod_{m=2}^{M}\|\mathbf{W}_m\|_2 \\
&\le (1+\tfrac{1}{M})^M\|\tilde{\boldsymbol{X}} - \hat{\boldsymbol{X}}\|F_k + (1+\tfrac{1}{M})^{M-1}\tfrac{\|\mathbf{U}_1^k\|_2}{\|\mathbf{W}_1^k\|_2}\|\tilde{\boldsymbol{X}}\|F_k \\
&\quad + \sum_{m=0}^{M-2}(1+\tfrac{1}{M})^m\tfrac{\|\mathbf{U}_{M-m}\|_2}{\|\mathbf{W}_{M-m}\|_2}\|\tilde{\boldsymbol{X}}\|F_k \\
&\le eBF_k\left(\tfrac{\|\mathbf{U}_1^k\|_2}{\|\mathbf{W}_1^k\|_2} + \sum_{m=2}^{M}\tfrac{\|\mathbf{U}_m\|_2}{\|\mathbf{W}_m\|_2}\right)
\end{aligned}
$$

where $F_k = \|\mathbf{W}_1^k\|_2\prod_{m=2}^{M}\|\mathbf{W}_m\|_2$, the last inequality is achieved by $(1+\tfrac{1}{m})^M \le e$ for $m \le M$, and $\|\tilde{\boldsymbol{X}}\| \le B$ in Assumption 1. Then $\mathbb{E}_{\boldsymbol{u}}\big[\|f_{\Theta_{\boldsymbol{u}}}(\tilde{\boldsymbol{X}}) - f_\Theta(\tilde{\boldsymbol{X}})\|^2\big]$ is given as

$$
\begin{aligned}
\sum_{k=1}^{d+1}\mathbb{E}_{\boldsymbol{u}}\big[\Delta_k^M\big] &\le \sigma reB\sum_{k=1}^{d+1}F_k\left(\|\mathbf{W}_1^k\|_2^{-1} + \sum_{m=2}^{M}\|\mathbf{W}_m\|_2^{-1}\right) \\
&\le \sigma reB\sum_{k=1}^{d+1}\left(\prod_{m=2}^{M}\|\mathbf{W}_m\|_2 + \sum_{m=2}^{M}\tfrac{F_k}{\|\mathbf{W}_m\|_2}\right) \\
&\le \sigma reB(d+1)M\kappa^{M-1}
\end{aligned}
$$

where $r = \big[2\log(2eh)\big]^{1/2}$, and the first inequality is achieved bounding the spectral norm of the random matrices $\mathbf{U}$'s using random matrix theory (See Section 4.4 in [53]). Hence, setting $\sigma = (reB(d+1)M\kappa^{M-1})^{-1}\gamma$, then we have

$$
\max_{\tilde{\boldsymbol{X}}\in\mathcal{X}}\mathbb{E}_{\boldsymbol{u}}\big[\|f_{\Theta_{\boldsymbol{u}}}(\tilde{\boldsymbol{X}}) - f_\Theta(\tilde{\boldsymbol{X}})\|^2\big] < \gamma.
$$

Given any ReLU network satisfying the Assumptions 1 and 2 and with bounded spectral norm on its weights, we can upper bound its expected loss using the network sharpness, measured by some perturbations on the network parameters. $\qquad\square$

# B Synthetic details

In this section, we cover details regarding our synthetic data generation process and experiments. We first provide an overview of our data generation, and then we will cover a supplementary linear example.

## B.1 Synthetic data generating process

Here we describe our synthetic data generation process in detail. We enumerated all nodes in $G$ randomly. We generated random DAG instantiations with a randomly sampled branching factor up to the number of nodes in the DAG for our synthetic DAG generation. Edges were randomly added to the graph until either the branching factor was met or no more edges can be added without violating graphical acyclicity. We provide pseudocode for our synthetic DGP in Algorithm 1. For each random DAG in our experiment we randomly chose a $\sigma$ between 0.3 and 1, and we set $\mu = 0$ and $w = 1$.

For our experiments in the main paper, we use the following settings. In the linear case, each variable is equal to the sum of its parents plus noise. For the nonlinear case, each variable is equal to the sum of the sigmoid function of each parent plus noise.

---

**Algorithm 1** Synthetic Data Generating Process (DGP)

---

**Input:** A Graphical structure $G$, a mean $\mu$, standard deviation $\sigma$, edge weights $w$, a dataset size $n$.
**Output:** A dataset according to $G$ with $n$ samples.
**Function:** `gen_data`$(G, \mu, \sigma, w, n)$**:**
$e \leftarrow$ edges of $G$
$G_{sorted} \leftarrow$ `topological_graph_sort`$(G)$
$ret \leftarrow$ empty list
**for** $node \in G$ **do**
    Append to $ret[node]$ a list of Gaussian ($\mu$ and $\sigma$) randomly sampled list of size $n$.
**end for**
**for** $node \in G_{sorted}$ **do**
    **for** $par \in \{parents(node)\}$ **do**
        $ret[node]\ +=\ ret[par] * w(par, node)$, where $w(par, node)$ is the edge weight from $par$
        to $node$. Note that a non-linear function can be applied to $ret[par]$ to convert this into a
        non-linear data generator.
    **end for**
**end for**
**return** $ret$.

---

## B.2 Experiments on linear toy example

Table 4: Comparison of benchmark regularizers in terms of MSE ($\pm$ standard deviation) for linear synthetic datasets of size $n$ generated according to Fig. 1 using 10-fold cross-validation. A held-out test set of 1000 samples was generated and used for evaluating each method. Bold denotes best performing regularizer.

| Regularizer | $n = 500$ | $n = 1000$ | $n = 5000$ | $n = 10000$ | $n = 50000$ |
|---|---|---|---|---|---|
| L1 | $1.413 \pm 0.091$ | $1.210 \pm 0.045$ | $1.051 \pm 0.009$ | $1.022 \pm 0.005$ | $1.004 \pm 0.006$ |
| L2 | $1.327 \pm 0.057$ | $1.208 \pm 0.064$ | $1.027 \pm 0.009$ | $1.022 \pm 0.008$ | $1.008 \pm 0.004$ |
| Dropout (0.2) | $1.287 \pm 0.045$ | $1.237 \pm 0.011$ | $1.216 \pm 0.003$ | $1.213 \pm 0.004$ | $1.209 \pm 0.002$ |
| Dropout (0.5) | $1.227 \pm 0.036$ | $1.191 \pm 0.017$ | $1.159 \pm 0.003$ | $1.161 \pm 0.006$ | $1.157 \pm 0.006$ |
| SAE | $1.323 \pm 0.152$ | $1.164 \pm 0.033$ | $1.138 \pm 0.026$ | $1.137 \pm 0.009$ | $1.145 \pm 0.019$ |
| Batch Norm | $1.470 \pm 0.056$ | $1.320 \pm 0.055$ | $1.095 \pm 0.020$ | $1.044 \pm 0.009$ | $1.018 \pm 0.009$ |
| Input Noise | $1.340 \pm 0.068$ | $1.268 \pm 0.034$ | $1.089 \pm 0.020$ | $1.049 \pm 0.011$ | $1.017 \pm 0.009$ |
| MixUP | $1.306 \pm 0.074$ | $1.214 \pm 0.040$ | $1.112 \pm 0.012$ | $1.075 \pm 0.008$ | $1.047 \pm 0.006$ |
| CASTLE | $\mathbf{1.205 \pm 0.093}$ | $\mathbf{1.042 \pm 0.024}$ | $\mathbf{1.009 \pm 0.018}$ | $\mathbf{1.008 \pm 0.015}$ | $\mathbf{1.004 \pm 0.018}$ |

Using our linear method, we performed experiments on our toy example in Figure 1. We use the same experimental setup from the toy example in the main manuscript but with linear settings. Our

results are shown in Table 4, which demonstrates that CASTLE is the superior regularizer over all dataset sizes (similar to the nonlinear case).

## C Supplementary experiments, details, and results

In this section, we provide additional experiments to supplement the main manuscript.

### C.1 Sensitivity analysis and hyperparameter optimization

Before we present further results, we first provide a sensitivity analysis on $\lambda$ from (5). We use our synthetic DGP to synthesize a random DAG with between 10 and 150 nodes. We generated 2000 test samples and a training set with between 1000 and 5000 samples. We repeated this 50 times. Using 10-fold cross-validation we show a sensitivity analysis over $\lambda \in \{0.01, 0.1, 1, 10, 100\}$ in Figure 4 in terms of average rank. We compare using average rank since each experimental run (random DAG) will vary significantly in the magnitude of errors. Based on these results, for all of our experiments in this paper we use $\lambda = 1$, i.e., $\log(\lambda) = 0$. After fixing $\lambda$, our model has only one hyperparameter $\beta$ to tune. For $\beta$ in (6), we performed a standard grid search for the hyperparameter $\beta \in \{0.001, 0.01, 0.1, 1\}$.

Figure 4: Sensitivity analysis on $\lambda$.

### C.2 Scalability analysis

We perform an analysis of the scalability of CASTLE. Using our synthetic DAG and dataset generator, we synthesized datasets of 1000 samples. We used the same experimental setup used for the synthetic experiments. We present the computational timing results for CASTLE as we increase the number of input features on inference and training time in Figure 5. We see that the time to train 1000 samples grows exponentially with the feature size; however, the inference time remains linear as expected. Inference time on 1000 samples with 400 features takes approximately 2 seconds, while training time takes nearly 70 seconds. Computational time scales linearly with increasing the number of input samples. Experiments were conducted on an Ubuntu 18.04 OS using 6 Intel i7-6850K CPUs.

Figure 5: CASTLE scalability analysis

### C.3 Additional results

(a) Regression

(b) Classification

Figure 6: Comparison of CASTLE against benchmark regularization methods in terms or average rank across each fold (10-fold cross-validation) for regression (a) and classification (b) tasks. For clarity, we have sorted the datasets by average rank of CASTLE in decreasing order. In comparison to the other benchmarks, CASTLE maintains stable performance across datasets. Higher rank is better.

Table 5: Complete table of benchmark regularizers on regression in terms of test MSE ($\pm$ standard deviation) for experiments on real datasets using 10-fold cross-validation. Bold denotes lowest test MSE. For readability we split the table into two.

| $\mathcal{D}$ | Baseline | L1 | L2 | Dropout 0.2 | Dropout 0.5 |
|---|---|---|---|---|---|
| BH | $0.141 \pm 0.023$ | $0.137 \pm 0.025$ | $0.131 \pm 0.014$ | $0.168 \pm 0.032$ | $0.389 \pm 0.106$ |
| WQ | $0.747 \pm 0.038$ | $0.747 \pm 0.043$ | $0.746 \pm 0.039$ | $0.738 \pm 0.029$ | $0.850 \pm 0.068$ |
| FB | $0.758 \pm 1.017$ | $0.663 \pm 0.796$ | $1.341 \pm 1.069$ | $0.429 \pm 0.449$ | $0.597 \pm 0.313$ |
| BC | $0.359 \pm 0.061$ | $0.342 \pm 0.037$ | $0.370 \pm 0.142$ | $0.334 \pm 0.030$ | $0.434 \pm 0.080$ |
| SP | $0.416 \pm 0.108$ | $0.421 \pm 0.181$ | $0.550 \pm 0.291$ | $0.285 \pm 0.042$ | $0.482 \pm 0.128$ |
| CM | $0.536 \pm 0.103$ | $0.574 \pm 0.125$ | $0.527 \pm 0.060$ | $0.327 \pm 0.025$ | $0.519 \pm 0.064$ |
| ME | $0.885 \pm 0.056$ | $0.878 \pm 0.062$ | $0.935 \pm 0.060$ | $0.729 \pm 0.032$ | $0.710 \pm 0.022$ |

| $\mathcal{D}$ | SAE | Batch Norm | Input Noise | MixUp | CASTLE |
|---|---|---|---|---|---|
| BH | $0.148 \pm 0.027$ | $0.139 \pm 0.021$ | $0.137 \pm 0.018$ | $0.194 \pm 0.064$ | $\mathbf{0.123 \pm 0.016}$ |
| WQ | $0.727 \pm 0.030$ | $0.723 \pm 0.039$ | $0.771 \pm 0.036$ | $0.712 \pm 0.018$ | $\mathbf{0.708 \pm 0.030}$ |
| FB | $0.372 \pm 0.168$ | $0.705 \pm 0.396$ | $0.609 \pm 0.511$ | $0.385 \pm 0.208$ | $\mathbf{0.246 \pm 0.153}$ |
| BC | $0.322 \pm 0.021$ | $0.325 \pm 0.024$ | $0.319 \pm 0.022$ | $0.322 \pm 0.030$ | $\mathbf{0.318 \pm 0.036}$ |
| SP | $0.228 \pm 0.022$ | $0.318 \pm 0.062$ | $0.389 \pm 0.095$ | $0.267 \pm 0.072$ | $\mathbf{0.200 \pm 0.020}$ |
| CM | $0.387 \pm 0.034$ | $0.470 \pm 0.047$ | $0.495 \pm 0.081$ | $0.376 \pm 0.030$ | $\mathbf{0.326 \pm 0.031}$ |
| ME | $0.800 \pm 0.046$ | $0.892 \pm 0.096$ | $0.855 \pm 0.042$ | $0.866 \pm 0.068$ | $\mathbf{0.694 \pm 0.023}$ |

Table 6: Comparison of benchmark regularizers on classification in terms of test AUROC ($\pm$ standard deviation) for experiments on real datasets using 10-fold cross-validation. Bold denotes highest test test AUROC. For readability we split the table into two.

| $\mathcal{D}$ | Baseline | L1 | L2 | Dropout 0.2 | Dropout 0.5 |
|---|---|---|---|---|---|
| CC | $0.764 \pm 0.009$ | $0.766 \pm 0.007$ | $0.768 \pm 0.010$ | $0.776 \pm 0.009$ | $0.756 \pm 0.023$ |
| PD | $0.799 \pm 0.008$ | $0.793 \pm 0.013$ | $0.788 \pm 0.024$ | $0.797 \pm 0.010$ | $0.800 \pm 0.012$ |
| BC | $0.721 \pm 0.018$ | $0.726 \pm 0.011$ | $0.712 \pm 0.045$ | $0.713 \pm 0.022$ | $0.718 \pm 0.024$ |
| LV | $0.559 \pm 0.061$ | $0.594 \pm 0.020$ | $0.586 \pm 0.028$ | $0.579 \pm 0.053$ | $0.580 \pm 0.033$ |
| SH | $0.915 \pm 0.015$ | $0.921 \pm 0.006$ | $0.919 \pm 0.011$ | $0.922 \pm 0.017$ | $0.914 \pm 0.010$ |
| RP | $0.782 \pm 0.071$ | $0.801 \pm 0.013$ | $0.796 \pm 0.013$ | $0.743 \pm 0.052$ | $0.721 \pm 0.057$ |
| MG | $0.644 \pm 0.109$ | $0.675 \pm 0.114$ | $0.647 \pm 0.134$ | $0.628 \pm 0.081$ | $0.679 \pm 0.090$ |

| $\mathcal{D}$ | SAE | Batch Norm | Input Noise | MixUp | CASTLE |
|---|---|---|---|---|---|
| CC | $0.774 \pm 0.012$ | $0.773 \pm 0.009$ | $0.772 \pm 0.012$ | $0.778 \pm 0.009$ | $\mathbf{0.787 \pm 0.007}$ |
| PD | $0.796 \pm 0.010$ | $0.773 \pm 0.024$ | $0.796 \pm 0.013$ | $0.802 \pm 0.016$ | $\mathbf{0.817 \pm 0.004}$ |
| BC | $0.605 \pm 0.068$ | $0.727 \pm 0.012$ | $0.722 \pm 0.026$ | $0.700 \pm 0.055$ | $\mathbf{0.731 \pm 0.010}$ |
| LV | $0.542 \pm 0.095$ | $0.583 \pm 0.026$ | $0.597 \pm 0.041$ | $0.553 \pm 0.092$ | $\mathbf{0.595 \pm 0.032}$ |
| SH | $0.701 \pm 0.205$ | $0.913 \pm 0.013$ | $0.922 \pm 0.005$ | $0.921 \pm 0.005$ | $\mathbf{0.929 \pm 0.007}$ |
| RP | $0.774 \pm 0.103$ | $0.802 \pm 0.018$ | $0.796 \pm 0.009$ | $0.730 \pm 0.043$ | $\mathbf{0.814 \pm 0.014}$ |
| MG | $0.597 \pm 0.135$ | $0.672 \pm 0.072$ | $0.671 \pm 0.138$ | $0.685 \pm 0.081$ | $\mathbf{0.731 \pm 0.036}$ |

In this subsection, we provide supplementary results on real data. In addition to the public datasets in the main paper, we provide experiments on some additional datasets. Specifically, we perform experiments on the Medical Expenditure Panel Survey (MEPS) [54]. This dataset contains samples from a broad survey of families and individuals, their medical providers, and employers across the US. MEPS is mainly concerned with collecting data related to health service utilization, frequency, cost, payment, and insurance coverage for Americans. For this dataset, we predicted health service utilization. We abbreviate MEPS as ME. We also provided additional experimentation on the Meta-analysis Global Group in Chronic heart failure database (MAGGIC), which holds data for 46,817 patients gathered from 30 independent clinical studies or registries [55]. For this dataset, we predicted mortality in patients with heart failure. We abbreviate MAGGIC as MG in Table 6.

We provide regression results on real data in Table 5. We provide classification results on real data in Table 6. Lastly, we depict the regression and classification results highlighted in the main paper in terms of rank. In Figure 6, we see that for both regression and classification, CASTLE performs the best, and there is no definitive runner-up benchmark method testifying to the stability of CASTLE as a reliable regularizer.

## C.4 CASTLE ablation study

We provide an ablation study on CASTLE to understand the sources of gain of our methodology. Here we execute this experiment on our real datasets used in the main manuscript. We show the results of our ablation on our CASTLE regularizer to highlight our sources of gain in Table 7.

Table 7: Ablation study of CASTLE on real datasets to highlight sources of gain.

| Dataset | $\mathcal{L}_N(f_\Theta) + \mathcal{V}_{\Theta_1}$ | $\mathcal{R}_{\Theta_1} + \mathcal{V}_{\Theta_1}$ | $\mathcal{L}_N(f_\Theta) + \mathcal{R}_{\Theta_1}$ | $\mathcal{L}_N(f_\Theta) + \mathcal{R}_{\Theta_1} + \mathcal{V}_{\Theta_1}$ |
|---|---|---|---|---|
| | | Regression (MSE) | | |
| BH | $0.162 \pm 0.018$ | $0.226 \pm 0.158$ | $0.174 \pm 0.025$ | $\mathbf{0.123 \pm 0.016}$ |
| WQ | $0.711 \pm 0.035$ | $0.753 \pm 0.013$ | $0.713 \pm 0.019$ | $\mathbf{0.708 \pm 0.030}$ |
| FB | $0.265 \pm 0.045$ | $0.327 \pm 0.088$ | $0.451 \pm 0.032$ | $\mathbf{0.246 \pm 0.150}$ |
| BC | $0.362 \pm 0.040$ | $0.416 \pm 0.009$ | $0.373 \pm 0.016$ | $\mathbf{0.318 \pm 0.036}$ |
| SP | $0.338 \pm 0.181$ | $0.212 \pm 0.018$ | $0.572 \pm 0.340$ | $\mathbf{0.200 \pm 0.020}$ |
| CM | $0.347 \pm 0.016$ | $0.334 \pm 0.007$ | $0.478 \pm 0.078$ | $\mathbf{0.326 \pm 0.031}$ |
| | | Classification (AUROC) | | |
| CC | $0.778 \pm 0.006$ | $0.780 \pm 0.008$ | $0.768 \pm 0.011$ | $\mathbf{0.787 \pm 0.007}$ |
| PD | $0.795 \pm 0.012$ | $0.792 \pm 0.012$ | $0.766 \pm 0.012$ | $\mathbf{0.817 \pm 0.004}$ |
| BC | $0.712 \pm 0.018$ | $0.722 \pm 0.008$ | $0.712 \pm 0.020$ | $\mathbf{0.731 \pm 0.010}$ |
| LV | $0.562 \pm 0.033$ | $0.586 \pm 0.023$ | $0.566 \pm 0.027$ | $\mathbf{0.595 \pm 0.032}$ |
| SH | $0.895 \pm 0.006$ | $0.889 \pm 0.011$ | $0.890 \pm 0.010$ | $\mathbf{0.929 \pm 0.007}$ |
| RP | $0.801 \pm 0.012$ | $0.802 \pm 0.014$ | $0.791 \pm 0.012$ | $\mathbf{0.814 \pm 0.014}$ |

## C.5 Weight characterization

In this subsection, we provide a characterization of the input weights that are learned during the CASTLE regularization. We performed synthetic experiments using the same setup for generating Figure 3. We investigated two different scenarios. In the first scenario, we randomly generated DAGs where the target must have causal parents. We examine the average weight value of the learned DAG adjacency matrix in comparison to the truth adjacency matrix for the parents, children, spouses, and siblings of the target variable. The results are shown in Figure 7. As expected, the results show that when causal parents exist, CASTLE prefers to predict in the causal direction, rather than the anti-causal direction (from children).

Figure 7: Weight values on synthetic data when true causal structure is known. Our method favors using the parents of the target when available.

As a secondary experiment, we ran the same sets of experiments, except for DAGs without parents of the target variable. Results are shown in Figure 8. The results show that when parents are not available that CASTLE finds the children as predictors rather than spouses. Note that in this experiment, there will be no siblings of the target variable, since the target variable has no parents.

Lastly, CASTLE does not reconstruct features that do not have causal neighbors in the discovered DAG. To highlight this, in our noise variable experiment, we show the average weighting of the input layers. In the right-most figures of Figure 7 and Figure 8, it is evident that the weighting is much lower (near zero) for the noise variables in comparison to the other variables in the DAG. This highlights the advantages of CASTLE over SAE, which naively reconstructs all variables.

Figure 8: Weight values on synthetic data when true causal structure is known. This simulation was run with target variables not having any causal parents (and therefore no siblings as well). Our method favors using the children rather than spouses of the target.

## C.6 Dataset details

In Table 8, we provide details of the real world datasets used in this paper. We demonstrated improved performance by CASTLE across a diverse collection of datasets in terms of sample and feature size.

Table 8: Real-world dataset details.

| Dataset | Sample size | Feature size |
| --- | --- | --- |
| Boston Housing (BH) | 506 | 14 |
| Wine Quality (WQ) | 4894 | 12 |
| Facebook Metrics (FB) | 500 | 19 |
| Bioconcentration (BC) | 779 | 14 |
| Student Performance (SP) | 649 | 33 |
| Community and Crime (CM) | 1994 | 128 |
| Contraceptive Choice (CC) | 1472 | 9 |
| Pima Diabetes (PD) | 768 | 9 |
| Las Vegas Ratings (LV) | 504 | 20 |
| Statlog Heart (SH) | 270 | 13 |
| Retinopathy (RP) | 1151 | 20 |
| Medical Expenditure Panel Survey (ME) | 15786 | 139 |
| Meta-analysis Global Group in Chronic (MG) | 40367 | 33 |

## Additional References for Appendices

[54] Agency for Healthcare Research and Quality. Medical expenditure panel survey (meps), 2020.

[55] Chih M. Wong et al. Heart failure in younger patients: the Meta-analysis Global Group in Chronic Heart Failure (MAGGIC). *European Heart Journal*, 35(39):2714–2721, 06 2014.