[Reviews · NeurIPS 2020]

Review 1

Summary and Contributions: The proposed work introduces a regularization method for improving generalization performance by learning a causal graph. The proposed work also discovers the optimal features based on the topology of the causal graph that also acts as an auxiliary task for learning a predictive model. This is in contrast to methods that use reconstruction loss of all the features as a regularizer and force the model to reconstruct features regardless of any direct dependency in the discovered DAG. --- After reading the author response, I found the additional baselines to be compelling experimental evidence, so I weakly recommend acceptance.

Strengths: The paper is very clearly written. The proposed work uses the idea of DAG with NO TEARS in a feature space of a neural network. As far as the reviewer is aware, this has not been tried before. The overall idea is not new, but the application of existing idea is new.

Weaknesses: The empirical results seem a bit weak. In Table 2, it's not obvious to me, why the performance with BN, input noise as well as weight noise is worse as compared to the baseline. Similarly, in table 3, the proposed method marginally improves the performance as compared to classification. It would also be interesting the effect of trying popular regularizers in the case of non-linear networks like mixup [1] and manifold mixup [2]. [1] Mixup https://arxiv.org/abs/1710.09412 [2] Manifold Mixup https://arxiv.org/abs/1806.05236

Correctness: Yes. Claims and method seems correct.

Clarity: Yes, the paper is well written.

Relation to Prior Work: Yes, the relevant work is very well discussed.

Reproducibility: Yes

Additional Feedback:


Review 2

Summary and Contributions: This paper proposes a CASTLE as a regularizer that aims at enforcing the input vertices to form a DAG and reconstruct the nodes in the Markov Blanket of the output. This method can be extended to MLP. A generalization bound is derived to guarantee that the method can achieve acceptable out-of-sample generalization. Experiments are applied to the synthetic dataset and real-world datasets and can show improvement compared with other regularization methods.

Strengths: This work assumes that the covariates and the response variable are generated in a DAG. Correspondingly, the authors proposed to introduce R(W) as a regularizer to enforce the learned weight coefficients of covariates and the response variable to form a DAG. Besides, compared with previous work that leverages reconstruction loss as a regularizer, the authors further introduce V(W) as a regularizer to enforce that only the nodes that are related to Y (in the sense of Markov Blanket), which can further avoid over-fitting. In a nutshell, the propose method aligns well with the motivation which is reasonable. Besides, the authors give a theoretical analysis of out-of-sample generalization bound. The experimental results can support the claims and show the superority of the proposed methods, in term of out-of-sample generalization.

Weaknesses: This work does not introduce what is the form of V(W), is it \ell-1 type sparsity? What's more, this work lacks theoretical analysis and experimental results to support that such a regularizer can identify the Markov Blanket of Y, i.e., the model selection consistency, at least in linear cases. Besides, in many realistic applications such as image classification, the covariates are sensory-level hence it may not be reasonable to assume a DAG among these covariates and the response variable. It would be better if the work can stand on a more general assumption.

Correctness: Yes.

Clarity: Yes.

Relation to Prior Work: Yes.

Reproducibility: Yes

Additional Feedback:


Review 3

Summary and Contributions: The authors propose a regularization method (CASTLE) based on learned causality relationship among input variables. The proposed method first learns a DAG representing the causality relationship and then reconstructs each input variable using its parents in DAG.

Strengths: This paper propose a novel regularization method based on causal DAG. Rather than using prior knowledge to construct the DAG, the author propose to learn the DAG which makes the method flexible and generalizable. In addition, the author has shown extensive efforts in evaluating the proposed method against other regularization baselines in many public datasets.

Weaknesses: This work suffers from several major weaknesses. First of all, the writing and organization of this paper is disappointing. The writing quality poses great difficulty for readers to understand the paper. Moreover, several claims about the methodology are not well-supported. In addition, the experimental set-up is flawed and needs further improvement.

Correctness: There exist some flaws in the experimental setups: - The search space for the hyper-parameters of the baseline method is too limited. For instance, a grid search on three lambda value for L1 would be too small to address the variance in scale for the public datasets. Perhaps a better way is to perform a random search for K hyper-param settings and report the mean performance for the top k models (k<<K). - The learning rate, which could significantly affect model performance, is not tuned for baseline methods and the proposed model. - While the authors include L2 regularization in syntetic data experiment, it's missing from Table 2. Regarding the methodology: - In line 160 W is defined as an adjacency matrix. In the same section, W is also used as weights associated with input variables which have negative values. Does the two definitions of W conflict with each other? - The authors mention in the introduction section that CASTLE only reconstructs variables using its parents in the causal DAG. It's unclear how this constraint is enforced in the paper. - In line 116, what are u? Hidden confounding factors? - In line 86-87, the authors claim that reconstructing using only neighbors in causal DAG is more favorable than using all variables. Why is that? - In line 146-147, the authors claim that siblings in causal DAG share learnable similarities. It would be better to give some example of shared similarities and support this claim.

Clarity: The writting of this paper needs to be significantly improved. While the paper has a clear structure, most of the contents are poorly organized and represented. - The usage of notation is chaotic. For instance, in line 157, \tilde(X) is defined both as collection of random variables and a N*(d+1) matrix. Such notation "overloading" is very confusing. - Line 162 introduces three regularization terms without explict description. - Line 172, what is layer size? Is it number of output units? - Line 165 introduces a theorem that is used to justify a regularization term. However, this theorem is never described in the paper.

Relation to Prior Work: The related work section is staisfactory. Perhaps the authors could have more discussion on the DAG discover methodlogy.

Reproducibility: No

Additional Feedback:


Review 4

Summary and Contributions: The aim of this paper is to improve performance of supervised learning on out-of-bag samples. In the case of deep networks, regularization helps mitigate overfit but does not exploit the structure of the feature variables and their relation to the outcome when the DGP can be represented by a causal DAG. The authors propose CASTLE, which jointly learns the causal graph while performing regularization. In particular, the adjacency matrix of the learned DAG is used in the input layers of neural network, which translates to the penalty function being decomposed into the reconstruction loss found in SAE, a (new) acyclicity loss, and a capacity-based regularizer of the adjacency matrices. Unlike other approaches, CASTLE improves upon capacity-based and auto-encoder-based regularization by exploiting the DAG structure for identification of causal predictors (parents of Y, if they exist) and for target selection for reconstruction regularization (features that have neighbours in the underlying DAG). The main contributions lie in (1) combining the results from SAEs in [12] with continuous optimization for DAG learning in [39], [40]; and (2) borrowing results from the PAC-Bayes literature to derive an upper bound on the expected reconstruction loss under the DGP. CASTLE regularization was tested in the case where the causal DAG is parameterized by an M-layer feed-forward neural network with ReLU activations and layer size h and is shown to outperform several benchmark regularizers on both synthetic and real world data sets.

Strengths: This work clearly lies at the intersection of causal discovery and machine learning. By learning the DGP the complexity of the FNN is reduced because the adjacency matrix is embedded in the input layer. The method relies on non-parametric SEMs as opposed to any particular parametric form (e.g. need not be exponential family). The theoretical results seem sound and are based on viewing the DAG regularizer as a prior for the (structure of) the input weight matrices. The empirical results are encouraging. One significance not stated is the gains in interpretability. While the overall goal is prediction, this approach is less black box than standard FNN because the choice of variables for feature selection or target selection are grounded in the structure of the learned causal DAG. I would imagine this to be a selling point to critics of black-box modelling. Further, exploiting recent results in continuous optimization for learning non-parametric DAGs makes the proposed method more feasible. This work has the potential to become a go-to regularizer, pending the scalability of the method. The main idea (exploiting causal structure in NN regularization) is a neat idea. Since it seems to readily take previous results in causal discovery and embed it in previous results on SAEs the results are not as novel per se, but that may be outweighed by the potential significance of the method.

Weaknesses: The main weakness is scalability and lack of available runtime information. There is no mention of how long it takes to run the algorithm. While the goal is improving out-of-bag prediction, what is the computational cost associated with this prediction improvement, particularly compared to SAEs? If the gains in OOB prediction error are minimal but the computational cost formidable, that is worth contemplating. On the other hand, since the adjacency matrix is embedded in the input layers' weight matrices, is there a computational gain over some of these other methods? Finally, the simulations looked at DAGs having 10 to 150 nodes, which can still be considered low for some applications. Again, does the computational burden of learning the causal structure outweigh the gains made in OOB prediction error?

Correctness: The theoretical claims and empirical evaluations appear sound.

Clarity: It appears that the authors are not always consistent with font usage. In Theorem 1, they use \cal{R}_{\Theta_1} in (8) but define $R_{\Theta_1}$ below. The former is referred to again below (8) on page 6 and in the appendix, but the latter is used on page 5. This is confusing. I'm assuming all instances refer to the same loss (acyclicity loss) but the notation needs to be corrected or clarified. While it is clear what how the DGP was generated for Table 2, it is not clear how the random DAG structures were generated for Figure 3. The DAG in Figure 1 looks rather easy to learn from a causal discovery perspective. To what extent was this challenged in the DAGs of Figure 3?

Relation to Prior Work: Yes.

Reproducibility: No

Additional Feedback: Authors need to correct the font issue mentioned above. Line 157: is X-tilde = [Y, X] or is it [X,Y] per Definition 1? Sentence 4 of the Appendix is not a complete sentence. ___________________ Post-rebuttal: I think the authors have adequately addressed several of the reviewers comments.

[Author Response · NeurIPS 2020]

We thank all the reviewers for their valuable suggestions and feedback. We kindly appeal to all the reviewers that they will reconsider and improve their scores, because as shown again in the new results provided in this rebuttal our method achieved the best performance in comparison to all the benchmarks across the real-world datasets. We believe our work presents a valuable and general regularization method for supervised learning models.

**[[Reviewer 1]]** ∎ **Strength of empirical results:** We would like to point out that the benefit of our method is not only (the magnitude of) the improved performance; we also show that our method is consistently providing the best regularization (across every dataset - both synthetic and real-data). For example, in the encircled portion of Figure A in this response, our method consistently achieved the best MSE in comparison to all the benchmarks across the real-world datasets. This consistency is not seen in the other benchmark regularizers, which exhibit higher variance in their average rank across all the datasets. This notion is also conveyed in the synthetic experiments in Figure 3 and for real-data (classification and regression) in Figure 5 in the original manuscript. ∎ **Experiments with Mixup:** We appreciate the recommendation for additional benchmarks. We have conducted the recommended experiments for Mixup and Manifold Mixup and show that CASTLE still outperforms the other benchmarks as shown in Table A (please compare with CASTLE in Table 3 in the original manuscript) and Figure A for the real datasets.

**[[Reviewer 2]]** ∎ **Notation clarification:** You are assuming correctly - $V(W)$ is the $\ell_1$ norm in our methodology. We will clarify this in the revised manuscript. ∎ **Generalization bounds:** The primary goal of our method is improving out-of-sample prediction performance which we use a generalization bound as justification. We did not prove the consistency of using a reconstruction loss and a norm-based regularizer in DAG learning which has already been proven in [49] and [50], respectively. ∎ **CNN Limitations:** We agree with you and will clarify the limitations of our method as we did for CNNs in the Broader Impact statement. ∎ **Additional results:** We reinforce the superiority of our proposed method by providing additional results in Table A and Figure A in the response to Reviewer 1.

**[[Reviewer 3]]** ∎ **Experimental hyperparameters:** As mentioned in lines 255-256, we performed a grid-search over a wide range of hyperparameters. We believe that this is fairly done for each benchmark, as we conducted the same grid-search for our model that we did for the other benchmark methods (lines 257-258) and applied early stopping for each. The performance gain from our regularizer is not due to improper hyperparameter tuning. ∎ **L2 missing:** Table 2 contains L2 regularization. ∎ **Definition of W:** We are not defining **W** twice. The adjacency matrix can be represented by a matrix containing negative values - see NOTEARS [39] and Non-parametric DAGs [40]. Because of this, we can embed the adjacency matrix in the input layers, **W**'s, of the proposed neural network (Section 3.3). ∎ **Prediction using causal parents:** Each feature is constructed using every other feature based on the DAG structure embedded in the neural network input layers. When a DAG is learned, the parent features (non-zero weights **W**) are obligated to construct each child (see Figure 2 and lines 166-168). ∎ **Variable $u$:** In Def. 1, each variable $u_i$ is specific for a feature $X_i$ in the DAG, they are not hidden confounders, but the random noise to generate the feature $X_i$. ∎ **Causal neighbors:** Consider the case where a variable is just noise, and therefore does not have any causally adjacent nodes (neighbors). Reconstruction methods, such as SAE, naively (and inefficiently) learns to reconstruct noise variables that have no causal implications on the target variable. Through DAG learning, our method does not reconstruct these variables as the input weight matrices get forced to zero (see Figures 3, 6, and 7). ∎ **Sibling variables:** We mean the function generating the sibling variables may share the some similarities. We will elaborate this point with concrete examples in the revised submission. ∎ **X vs $X$:** We define the random variables as $X$ and the corresponding data matrix as **X**. This is standard notation in machine learning. ∎ **Regularization terms:** The description of the regularization terms is given in lines 162-168. We will describe them in more detail in the revised paper. ∎ **Layer size:** We provided the layer size in Section 4 on line 264. ∎ **Acyclicity constraint:** Starting on line 165, we introduce and describe Theorem 1 from [39]. We describe Theorem 1 by saying, "the graph given by **W** is a DAG if and only if $R_{\mathbf{W}} = 0$." ∎ **Writing quality:** Although the other reviewers have positively acknowledged our exposition, we will work to improve the writing quality.

**[[Reviewer 4]]** ∎ **Scalability:** For a typical dataset with hundreds of features or less, the computational training time does not differ significantly between the regularizers. For example, on simulated data, an experimental run with 200 features, 2000 samples, and 200 epochs had an average training time of ∼55s and ∼64s for SAE and CASTLE, respectively, on an Intel i7-6850K CPU at 3.60GHz. We will incorporate a computational complexity analysis of our method as well as a demonstration of the computational trade-offs between our method and improved performance in the revised manuscript. ∎ **Notations:** We will correct the suggested typos in the revision.

Table A: Additional Experiments (real data).

| DATA | MIXUP | MIXUP-MAN |
|---|---|---|
| REGRESSION (MSE) | | |
| BH | 0.134 ± 0.019 | 0.130 ± 0.023 |
| WQ | 0.717 ± 0.030 | 0.712 ± 0.028 |
| FB | 0.329 ± 0.184 | 0.387 ± 0.316 |
| BC | 0.339 ± 0.025 | 0.325 ± 0.039 |
| SP | 0.208 ± 0.040 | 0.282 ± 0.023 |
| CM | 0.333 ± 0.025 | 0.386 ± 0.031 |
| CLASSIFICATION (AUROC) | | |
| CC | 0.763 ± 0.008 | 0.772 ± 0.007 |
| PD | 0.814 ± 0.018 | 0.808 ± 0.009 |
| BC | 0.719 ± 0.020 | 0.728 ± 0.013 |
| LV | 0.586 ± 0.041 | 0.571 ± 0.025 |
| SH | 0.904 ± 0.015 | 0.916 ± 0.011 |
| RP | 0.798 ± 0.016 | 0.804 ± 0.018 |

Figure A: Comparison in terms of average rank (in terms of MSE - lower is better). CASTLE has the best and most stable performance across all datasets.

[Meta-Review · NeurIPS 2020]

This paper proposes a new regularization method based on learning causal graphs. Three reviewers suggest accept, one indicates reject. All reviewers found that the method was extensively compared against proper baselines. The concerns about scalability were addressed by the rebuttal. R3 recommended reject based on concerns around correctness, which I found were adequately addressed in the rebuttal. Therefore, I recommend accept.